# Three-dimensional flagella structures from animals' closest unicellular relatives, the Choanoflagellates

Justine M Pinskey, Adhya Lagisetty, Long Gui, Nhan Phan, Evan Reetz, Amirrasoul Tavakoli, Gang Fu, Daniela Nicastro*

Department of Cell Biology, University of Texas Southwestern Medical Center, Dallas, United States

**Abstract** In most eukaryotic organisms, cilia and flagella perform a variety of life-sustaining roles related to environmental sensing and motility. Cryo-electron microscopy has provided considerable insight into the morphology and function of flagellar structures, but studies have been limited to less than a dozen of the millions of known eukaryotic species. Ultrastructural information is particularly lacking for unicellular organisms in the Opisthokonta clade, leaving a sizeable gap in our understanding of flagella evolution between unicellular species and multicellular metazoans (animals). Choanoflagellates are important aquatic heterotrophs, uniquely positioned within the opisthokonts as the metazoans' closest living unicellular relatives. We performed cryo-focused ion beam milling and cryo-electron tomography on flagella from the choanoflagellate species *Salpingoeca rosetta*. We show that the axonemal dyneins, radial spokes, and central pair complex in *S. rosetta* more closely resemble metazoan structures than those of unicellular organisms from other suprakingdoms. In addition, we describe unique features of *S. rosetta* flagella, including microtubule holes, microtubule inner proteins, and the flagellar vane: a fine, net-like extension that has been notoriously difficult to visualize using other methods. Furthermore, we report barb-like structures of unknown function on the extracellular surface of the flagellar membrane. Together, our findings provide new insights into choanoflagellate biology and flagella evolution between unicellular and multicellular opisthokonts.

*For correspondence:
daniela.nicastro@
utsouthwestern.edu

**Competing interest:** The authors declare that no competing interests exist.

## Editor's evaluation

This is a thorough, beautiful and compelling study of the flagellar structure of the choanoflagellate *S. rosetta* as reconstituted by cryo-electron tomography (now one of the handful of eukaryotic species whose flagella have been studied in such detail). The findings yield many important new insights of broad interest to the field (such as the similarity of outer dynein arm and radial spoke structure to metazoan cilia, the observation of a flagellar vane in that species, and the presence of mysterious barb structures).

## Introduction

Eukaryotic cilia and flagella (terms often used interchangeably) are long, microtubule-based structures that protrude from the cell surface. All major branches of the eukaryotic tree of life contain flagellated representatives, strongly suggesting the presence of one or more cilia or flagella in the last eukaryotic common ancestor (LECA) (*Cavalier-Smith, 2002*; *Mitchell, 2004*; *Mitchell, 2007*). The vast majority of eukaryotic life consists of unicellular organisms with flagella, which perform a variety of functions necessary for their survival, for example, aiding motility, feeding, avoiding predators, and sensing the environment (*Burki, 2014*; *Mitchell, 2007*). Multicellular eukaryotes, including animals (metazoans),

also rely on cilia and flagella for locomotion, developmental signaling, mucosal clearance, feeding, and reproduction. The structure of motile cilia and flagella is quite complex and contains several hundred different proteins (*Pazour et al., 2005*). Yet the mutation of a single flagellar protein can result in severe flagellar assembly or motility defects, which can lead to death or disease, including human ciliopathies (*Reiter and Leroux, 2017*).

Although the overall architecture of motile cilia and flagella is conserved, their protein structures, accessory features, and regulatory complexes also show some divergence throughout evolution. Most motile cilia and flagella contain a ring of nine outer doublet microtubules (DMTs) with a pair of central singlet microtubules, often referred to as the '9+2' arrangement (*Fawcett, 1954*), although exceptions exist, such as the vertebrate nodal cilia (9+0), eel sperm flagella (9+0) and rabbit posterior notochord cilia (9+4) (*Takeda and Narita, 2012*). The axonemal core in motile cilia and flagella contains 96 nm repeat units with two rows of dyneins, the outer and inner dynein arms (ODAs, IDAs), regulatory complexes like the nexin-dynein regulatory complex (N-DRC) and radial spokes (RSs), and the central pair complex (CPC), again with some exceptions (*Grossman-Haham et al., 2021*; *Gui et al., 2021*; *Han et al., 2022*; *Ma et al., 2019*; *Porter and Sale, 2000*; *Rao et al., 2021*; *Smith and Yang, 2004*; *Walton et al., 2021*). Despite these broad commonalities, ultrastructural studies have shown differences in the morphology of flagellar protein complexes (*Lin et al., 2014*; *Zheng et al., 2021*). Motile cilia and flagella also exhibit a variety of beating patterns including helical, planar, base to tip, tip to base, or reversible (*Blake and Sleigh, 1974*), and can be outfitted with an assortment of accessory structures, including mastigoneme hairs, paraflagellar rods, fibrous sheaths, outer dense fibers, and accessory microtubules (*de Souza and Souto-Padrón, 1980*; *Hyams, 1982*; *Irons and Clermont, 1982a*; *Irons and Clermont, 1982b*; *Mencarelli et al., 2008*; *Nakamura et al., 1996*; *Portman and Gull, 2010*; *Yubuki et al., 2016*).

Our understanding of flagellar ultrastructure and evolution is continually expanding through application of new technologies. Historically, much of our knowledge of flagellar architecture from diverse species has been based on conventional light and electron microscopy studies, which are inherently limited by detection limits and preservation artifacts. Protein sequence comparisons have also yielded important insights, particularly into dynein evolution in eukaryotic flagella (*Kollmar, 2016*), although this required manual annotation of thousands of genes from hundreds of species, not particularly sustainable for examining hundreds of flagellar proteins. Similarly, comparative proteomic studies have also largely contributed to our understanding of flagella composition and evolution (*Pazour et al., 2005*; *Sigg et al., 2017*), although both sequence comparisons and proteomics are limited in their ability to predict protein localization and interactions. As a result, our knowledge of detailed flagellar morphology, function, and evolution has remained restricted. Advances in cryo-electron tomography (cryo-ET) have enabled visualization of native flagellar structures with unparalleled resolution, enhancing our ability to compare flagellar morphology across species and make inferences about their evolution and function, although cilia and flagella from less than a dozen species have currently been examined using cryo-ET (*Carbajal-González et al., 2013*; *Fu et al., 2018*; *Lin et al., 2012a*; *Lin et al., 2012b*; *Lin and Nicastro, 2018*; *Lin et al., 2014*; *Nicastro et al., 2011*; *Pigino et al., 2012*).

The molecular structures of motile cilia and flagella from several multicellular animals (metazoans) have been studied using cryo-ET, but similar high-resolution structural information is lacking for unicellular organisms in the same Opisthokonta clade, preventing structural comparison between metazoans and their close unicellular relatives. Cryo-ET studies of unicellular species from other suprakingdoms, such as the Archaeplastida (e.g. *Chlamydomonas*), Alveolata (e.g. *Tetrahymena*), and Excavata (e.g. *Trypanosoma*), have revealed significant morphological differences between unicellular and multicellular motile cilia and flagella, including dynein number and arrangement, CPC shape, microtubule inner proteins, and radial spoke head morphology (*Carbajal-González et al., 2013*; *Imhof et al., 2019*; *Lin et al., 2014*; *Pigino et al., 2011*; *Pigino et al., 2012*). However, these clades are phylogenetically quite distant from metazoans, raising questions about when and how these differences arose throughout the evolutionary timescale (*Figure 1A*).

Choanoflagellates are unicellular (or colonial) organisms within the Opisthokonta branch that share a last common ancestor with metazoans (the urchoanozoan) more than 600 million years ago (*Carr et al., 2008*; *King, 2004*; *Ruiz-Trillo et al., 2008*; *Steenkamp et al., 2006*). Because of their unique phylogenetic position, choanoflagellates provide important information on the origin and evolution of multicellular organisms (*King, 2004*). Though low-resolution features of choanoflagellate flagella have

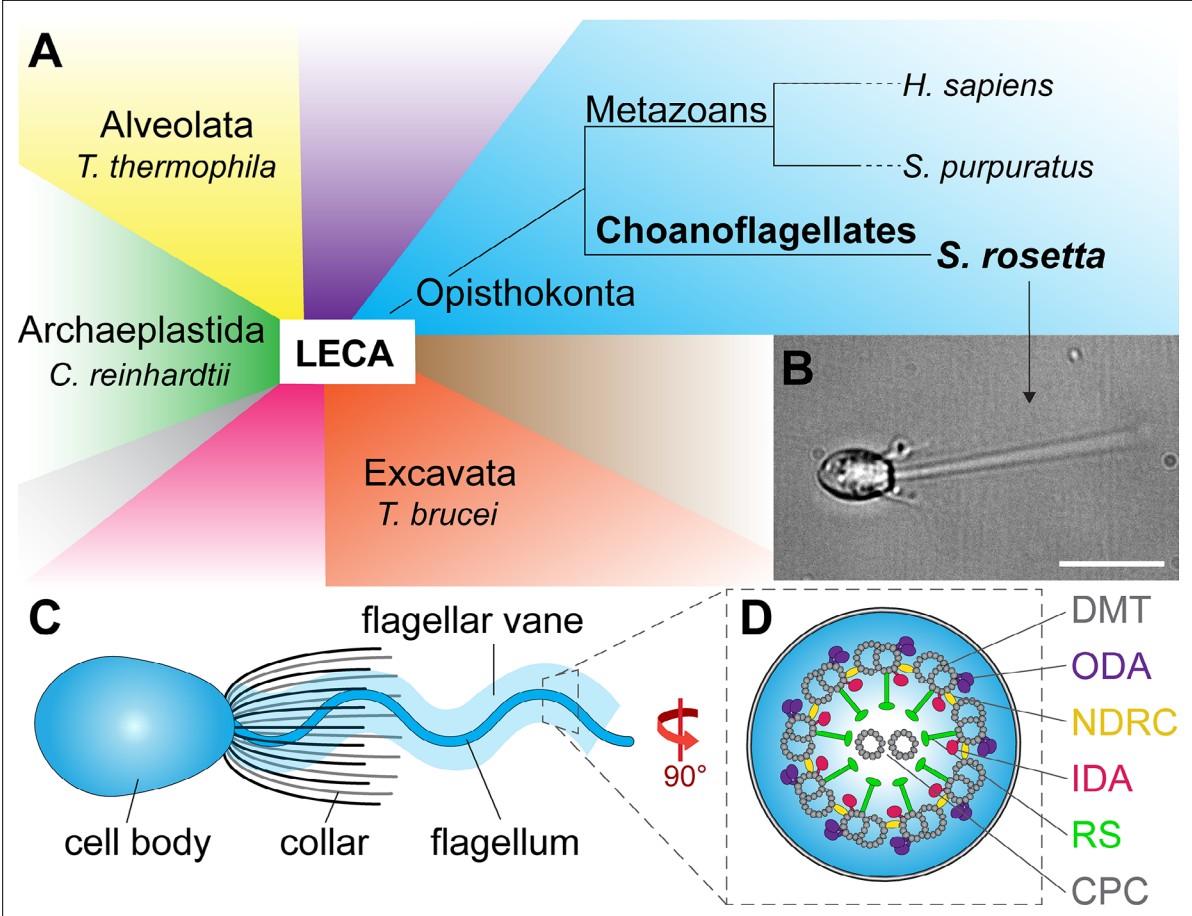

**Figure 1.** Phylogeny and flagellar features of the choanoflagellate *S.rosetta*. (**A**) Phylogenetic tree showing major eukaryotic suprakingdoms (colored blocks) stemming from the last common eukaryotic ancestor (LECA). Suprakingdoms with representatives that have been imaged using cryo-ET are labeled (i.e. Alveolata, Opisthokonta, Excavata, and Archaeplastida) with example species. Choanoflagellates are part of the Opisthokonta branch and form a sister group with metazoans, having shared a last common unicellular ancestor more than 600 million years ago. Whereas metazoans are multicellular animals, the choanoflagellates have remained unicellular/colonial. (**B**) Fixed *Salpingoeca rosetta* cell (a marine choanoflagellate). A short movie of an *S. rosetta* cell swimming and additional images of selected *S. rosetta* cell types can be found in *Figure 1—video 1* and *Figure 1— figure supplement 1*, respectively. (**C**) Overview cartoon of the choanoflagellate cell architecture, including the cell body and the ring of actin-based microvilli comprising the collar, which surrounds a single flagellum with a flagellar vane. (**D**) Cross-sectional diagram of the choanoflagellate flagellum indicating known flagellar components. The cross-section in this figure and throughout the paper are viewed from proximal towards the distal tip of the flagellum, and the longitudinal sections are shown with proximal on the left unless otherwise indicated. Labels: CPC, central pair complex; DMT, doublet microtubule; IDA and ODA, inner and outer dynein arm; N-DRC, nexin-dynein regulatory complex; RS, radial spoke. Scale bar: 10 µm (**B**).

The online version of this article includes the following video and figure supplement(s) for figure 1:

**Figure supplement 1.** Morphology of a subset of *S.rosetta* cell types.

**Figure 1—video 1.** Choanoflagellate cell swimming in culture imaged by light microscopy at ×40 magnification.

https://elifesciences.org/articles/78133/figures#fig1video1

been described (*Hibberd, 1975*; *Karpov, 2016*; *Leadbeater, 2014*), detailed molecular structures remain unexamined.

Here, we use cryo-focused ion beam milling (cryo-FIB) and cryo-ET to investigate the flagellum and other structures in the flagellar region of the marine choanoflagellate species *Salpingoeca rosetta*. Our tomographic reconstructions and 3D averages suggest that choanoflagellates and their metazoan relatives share similar morphology and arrangement of flagellar dyneins and their regulators, suggesting that these features were already present in the two groups' last common ancestor. Similarly, the *S. rosetta* CPC strongly resembles that of sea urchin (*Strongylocentrotus purpuratus*) sperm flagella. In contrast, however, we also observed flagellar features that appear to be unique to Choanoflagellates, such as previously unseen gaps and microtubule inner proteins (MIPs) in the DMTs, the flagellar vane,

which is a fine mesh of intertwined filaments extending bilaterally from the flagellar membrane, and barb-like structures, which protrude from the extracellular surface of the flagellar membrane. These findings expand our understanding of choanoflagellate biology and provide insights into the evolution of flagellar structures within the Opisthokonta branch.

## Results

*S. rosetta* cells contain a single flagellum, which extends from the cell body and is surrounded by a ring of 25–36 actin-based microvilli (*Figure 1B–D*, *Figure 1—video 1*, *Figure 1—figure supplement 1*; *Dayel et al., 2011*). As microbial filter feeders, choanoflagellates use the planar beat of their flagellum to generate both cell motility and microcurrents, which enable them to more easily engulf bacterial prey (*Pettitt et al., 2002*). The overall structure of the choanoflagellate flagellum has been previously studied using light and conventional electron microscopy techniques, revealing a 9+2 axonemal microtubule arrangement and a basal body that is surrounded by a microtubule rootlet structure (*Karpov, 2016*; *Karpov and Leadbeater, 1998*). We sought to visualize molecular structures within and surrounding the *S. rosetta* flagellum with improved resolution enabled by technical advances in cryo-FIB milling and cryo-ET imaging (*Marko et al., 2007*; *McIntosh et al., 2005*).

### Cryo-ET and subtomogram averaging facilitate high-resolution analyses of the *S. rosetta* flagellum

*S. rosetta* can transition between several cell types, including single-celled slow and fast swimmers, doublets, chains, rosettes, and thecate cells that attach to substrates through a secreted basal process (examples in *Figure 1—figure supplement 1*; *Dayel et al., 2011*). We rapidly froze starved choanoflagellate singlet cells in their slow- and fast-swimming morphological states. During plunge-freezing, areas close to the cell body were embedded in relatively thick ice (>500 nm); therefore, we used cryo-FIB milling to generate ~150–200 nm thin lamellae (sections) of the plunge-frozen cells before cryo-ET imaging (*Figure 2A-C*). In one cryo-FIB lamella, we captured part of the cell body with actin-filled collar microvilli extending outward and the proximal region of the flagellum from which we were able to record sequential cryo-tomograms along the flagellar length (*Figure 2D-F*). Within the reconstruction of the basal apparatus, we observe part of the basal body and the surrounding MTOC ring of dense material from which the lateral rootlet microtubules radiate outwards (*Figure 2E*; *Karpov, 2016*; *Pozdnyakov et al., 2017*). We observe multiple microvilli bases and many vesicles distributed throughout the apical end of the cell (*Figure 2E*). In addition, the flagellar vane filaments were clearly visible on two opposite sides of the flagellum and extended to the edges of the imaging area (~3 μm) (*Figure 2D and F*). Farther away from the cell body, the ice was sufficiently thin to perform cryo-ET imaging directly on the plunge-frozen flagella, where the 3D reconstructions also contained actin-based microvilli and thin vane filaments (*Figure 2G and H*).

To better resolve the molecular details of the *S. rosetta* flagellum, we performed subtomogram averaging of >7500 axonemal repeats (96 nm length) that were extracted from 54 cryo-tomograms (*Figure 2G*, green brackets; *Figure 3*), which yielded an average with 2.2 nm resolution (0.5 FSC criterion; *Figure 3—figure supplement 1*; *Table 1*). With this resolution, we observe that the axonemal repeats of *S. rosetta* flagella contain outer dynein arms with two motor domains each, the double-headed I1 (f) inner dynein complex, and six single-headed inner dynein arms, *a*, *b*, *c*, *e*, *g*, and *d* (*Figure 3*). Doublet-specific averages allowed us to identify the conserved bridge structures between DMTs 5 and 6 (*Afzelius, 1959*; *Lin et al., 2012b*). Similar to sea urchin sperm flagella (*Lin et al., 2012b*), the ODAs and a subset of IDAs (b, c, and e) on DMT 5 of the *S. rosetta* flagellum are replaced by the o-SUB5-6 and i-SUB5-6 structures (*Figure 3—figure supplement 2*; DMT5, green and orange arrowheads), thus allowing us to unambiguously determine the doublet numbers DMT1-9 within each reconstructed flagellum (*Figure 3—figure supplement 2*). We also observed a unique connection between the A-tubule and the base of IDA c on DMT 9, with a smaller partial density near the base of the A-tubule on DMT 1 (*Figure 3—figure supplement 2*, dark yellow arrowheads).

Most flagella contain three radial spokes per axonemal repeat (RS1-RS3), which project from the A-tubule toward the CPC, and regulate flagellar motility through poorly understood signaling mechanisms (*Zhu et al., 2017*). The *S. rosetta* flagellum also contains three full-length radial spokes per axonemal repeat (*Figure 3C, C', E, and F*) with somewhat variable radial spoke head positions,

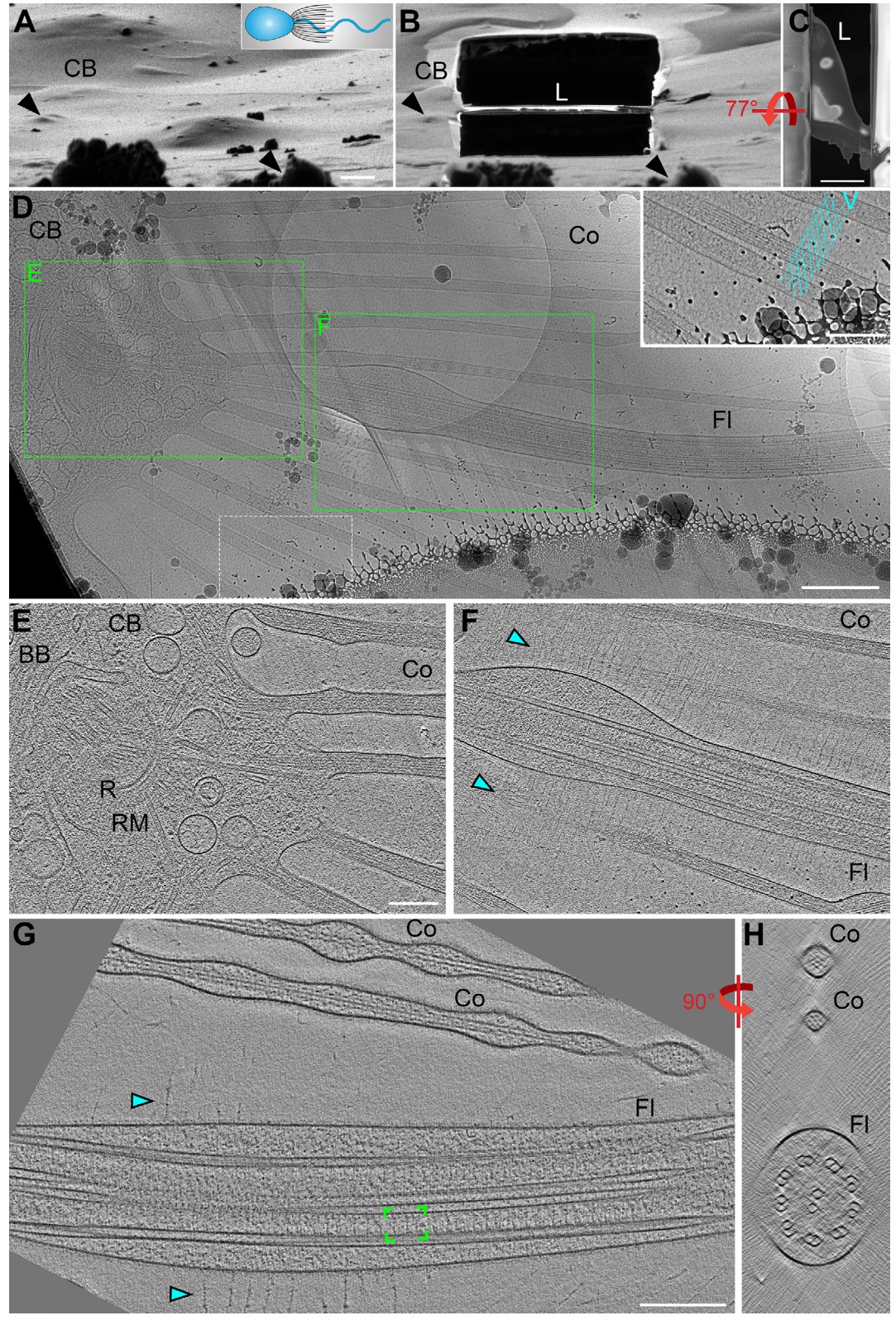

**Figure 2.** Cryo-FIB milling and cryo-ET enable visualization of flagellar structures. (**A,B**) Choanoflagellate cell before (**A**) and after (**B**) cryo-FIB milling, as viewed by the ion beam. The cartoon denotes the cell's orientation, with cell body (CB) to the left. Black arrowheads in A and B denote surface features in the ice to serve as landmarks for positional orientation. Note: the lamella (**L**) that includes the flagellum appears low relative to the cell body due to a visual illusion caused by the tilt and the several micron thick sputter/GIS-layer on top of the ice layer. (**C**) Perpendicular top view of the cryo-FIB milled

*Figure 2 continued on next page*

*Figure 2 continued*

lamella (shown in B) viewed with the electron beam. (**D**) Overview map of the milled flagellum (Fl), with green boxes indicating the positions of two sequential tomograms that were recorded from this lamella, shown in (**E and F**). The area within the white dashed line is magnified as an inset in the upper right corner, highlighting the regular meshwork of vane filaments which extend past the edges of the map. (**E–F**) Tomographic slices emphasizing the basal body (BB) and collar microvilli (Co) (**E**) and the proximal region of the flagellum (shown in F). Cyan arrowheads denote vane filaments. (**G–H**) Tomographic reconstruction of a whole (not cryo-FIB milled) *S. rosetta* flagellum in longitudinal (**G**) and cross-sectional (**H**) views. Green brackets indicate a single 96 nm axonemal repeat, thousands of which were used to generate the subtomogram averages shown in *Figure 3*. Other labels: R, ring of dense material (MTOC); RM, rootlet microtubules. Scale bars: 2 µm (**C**); 1 µm (A, applies also to B); 500 nm (**D**); 200 nm (D inset; E, applies also to F; G, applies also to H).

causing them to blur-out slightly in the averages (*Figure 3A and C*). This positional flexibility was likely because intact *S. rosetta* cells were frozen while their flagella were actively beating. To improve the resolution of the radial spoke heads, we performed local alignments focused on each of the three head domains (*Figure 3C'*). Similar to sea urchin and mammalian cilia and flagella, the shape of the *S. rosetta* radial spoke heads resemble narrow ice skates (*Figure 3C'and F*; *Figure 4*), rather than

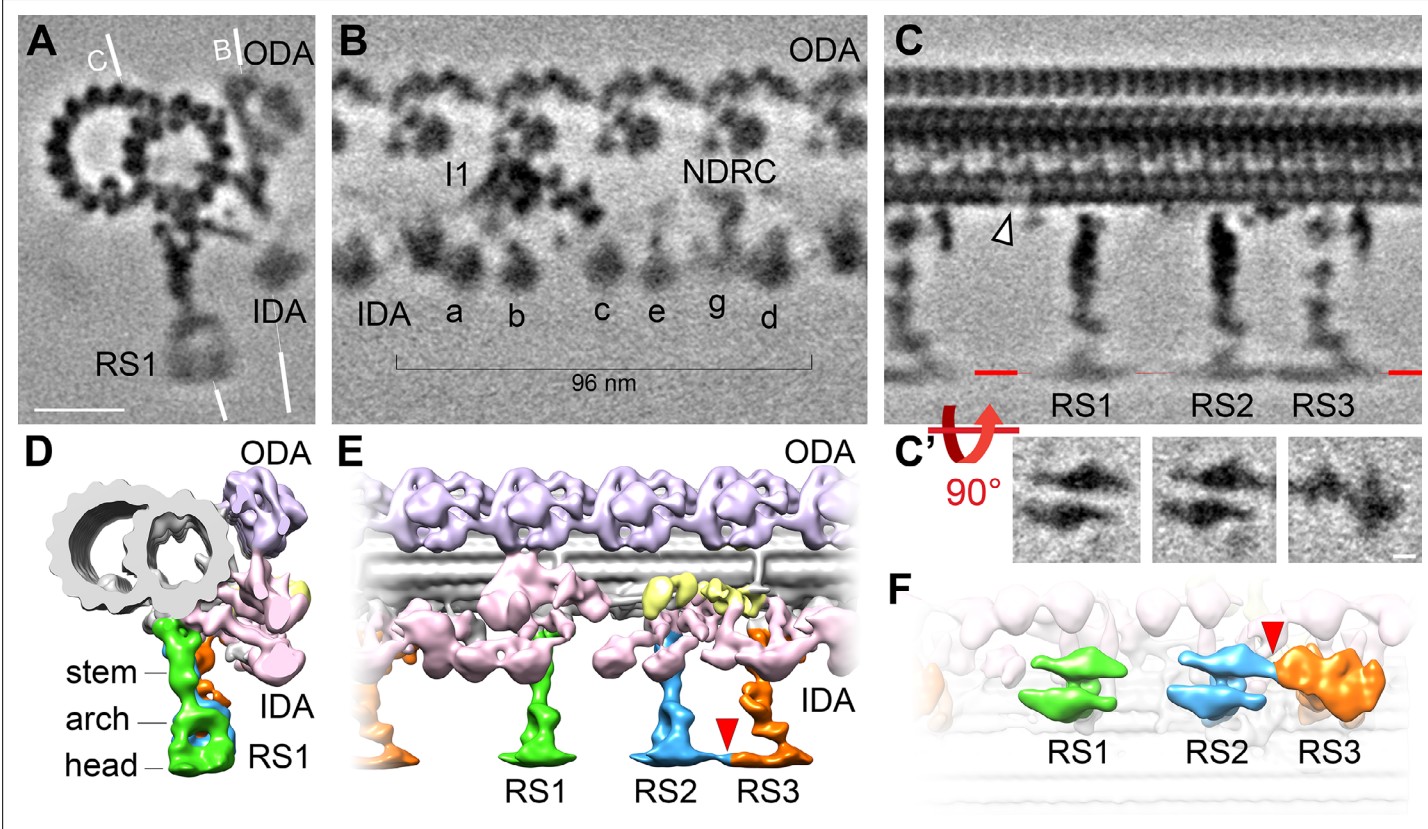

**Figure 3.** Cryo-ET of native choanoflagellate flagella reveals structural features of the 96 nm axonemal repeat. (**A–C**) Cross-sectional (**A**) and longitudinal (**B–C**) slices through the subtomogram average of the *S. rosetta* flagellar doublet microtubule. The white lines in (**A**) indicate the positions of the slices shown in (**B**) and (**C**). The white arrowhead in (**C**) denotes a hole in the A-tubule; some blurring appears because the hole was not present in all averaged repeats (see classification in *Figure 5*). Resolution information and tomogram/particle numbers are in *Figure 3—figure supplement 1* and *Table 1*. (**C'**) The radial spoke heads were blurred in the global subtomogram averages due to positional heterogeneity, therefore we performed local alignment refinements for each radial spoke head, which are displayed as viewed from the bottom. (**D–F**) Isosurface renderings of the averaged *S. rosetta* 96 nm axonemal repeat shown in cross-sectional (**D**), longitudinal (**E**), and bottom (**F**) views. *Figure 3—figure supplement 2* includes additional information on DMT-specific features. Labels: outer dynein arms (ODA, lavender), inner dynein arms (IDA, a-e, g, pink), I1 dynein (I1, pink), nexin-dynein regulatory complex (NDRC, yellow), and radial spokes (RS1, 2, and 3, green, blue, and orange, respectively). Scale bars: 20 nm (A, applies to A-C); 5 nm (applies to all panels in C').

The online version of this article includes the following figure supplement(s) for figure 3:

**Figure supplement 1.** Resolution of averaged *S. rosetta* flagellar structures.

**Figure supplement 2.** DMT-specific features in the *S.rosetta* flagellum.

**Table 1.** Summary of data included in this study.

| Specimen | Tomograms included | Averaged repeats | Resolution at 0.5 Fourier shell correlation criterion (nm) | Resolution at 0.143 Fourier shell correlation criterion (nm) | Used in Figure(s) |
| --- | --- | --- | --- | --- | --- |
| *S. rosetta* slow/fast swimmers* | 54 | 7584 | 2.2 | 1.8 | 3, 4, 5 |
| Central Pair Complex† | 28 | 1323 | 2.5 | 2.2 | 6 |
| Barb structures (with 4-fold symmetry)‡ | 17 | 600 | 2.5 | 2.2 | 7 |

*Resolution was estimated at the base of RS1 from a 64 voxel subvolume.
†Resolution was estimated at the central portion of the barb from a 64 voxel subvolume.
‡Resolution was estimated at C1a from a 32 voxel subvolume.

the broad radial spoke head morphology of other unicellular species like *Chlamydomonas* and *Tetrahymena* (*Figure 4*; *Barber et al., 2012*; *Grossman-Haham et al., 2021*; *Gui et al., 2021*; *Lin et al., 2014*; *Pigino et al., 2011*; *Pigino et al., 2012*; *Poghosyan et al., 2020*; *Zheng et al., 2021*). The head domains of *S. rosetta* RS1 and RS2 are separated from one another, whereas those of RS2 and RS3 are connected (*Figure 3C, E, and F*, *Figure 4*).

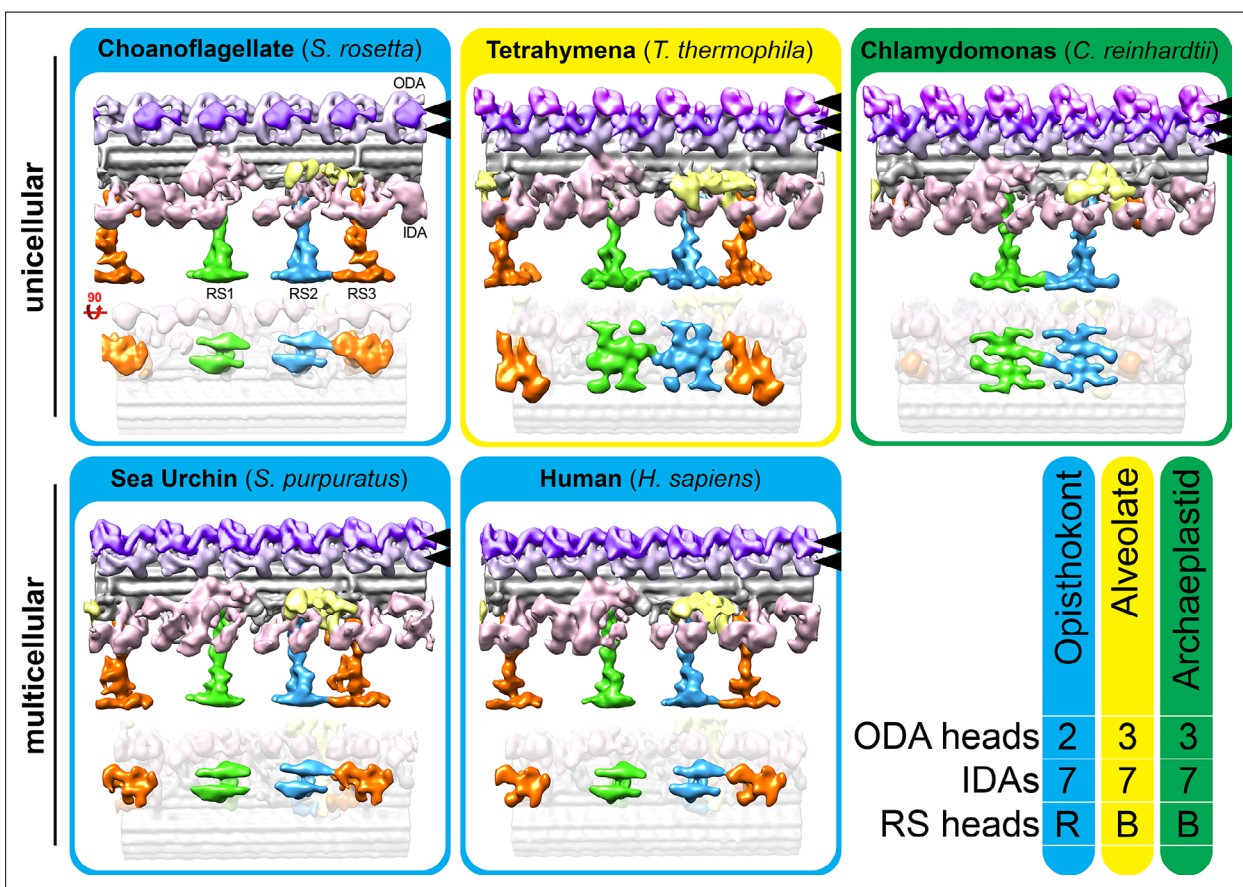

**Figure 4.** The flagellar structures of the unicellular choanoflagellate more closely resemble those of multicellular opisthokonts than unicellular organisms from other suprakingdoms. Isosurface renderings of the 96 nm flagellar repeats from unicellular (top row) vs. multicellular (metazoan; bottom row) species. The summary on the bottom right highlights that the flagella of opisthokonts, including the unicellular/colonial *S. rosetta*, contain two dynein heads per ODA and reduced (**R**) radial spoke (RS) heads, whereas *Tetrahymena* (Alveolata) and *Chlamydomonas* (Archaeplastida) contain three dynein heads per ODA and broad-shaped (**B**) RS heads. The dynein heads in each ODA are indicated with black arrowheads and are pseudocolored in pale and darker purple, and in magenta where a third dynein is present, to help distinguish between ODAs with two or three dynein heads. Each organism contained one double-headed and six single-headed inner dynein arms (IDAs). The averaged axonemal structures from species other than *S. rosetta* were previously published (*Lin et al., 2014*).

## *S. rosetta* doublet microtubules show conserved and unique features

Microtubule inner proteins (MIPs) are regularly distributed proteins that attach to the luminal side of flagellar microtubule walls (*Ichikawa et al., 2017*; *Kirima and Oiwa, 2018*; *Maheshwari et al., 2015*; *Nicastro et al., 2011*; *Nicastro et al., 2006*), and in other hyperstable microtubule species, including subpellicular microtubules in apicomplexan parasites (*Wang et al., 2021*) and ventral disc microtubules of *Giardia* (*Schwartz et al., 2012*). Many of the flagellar MIPs are highly conserved between species (*Ichikawa et al., 2017*; *Khalifa et al., 2020*; *Ma et al., 2019*; *Maheshwari et al., 2015*; *Nicastro et al., 2011*; *Nicastro et al., 2006*; *Song et al., 2020*), but some species-specific MIP features have also been reported, such as the *Chlamydomonas* beak-MIP (*Dymek et al., 2019*; *Hoops and Witman, 1983*), the *T. brucei*-specific B2, B4, B5, ponticulus MIPs, snake-MIP, ring MIP, and Ring-Associated MIP (RAM) (*Imhof et al., 2019*), and a connection of the B-tubule MIP3 to the mid-partition in *Tetrahymena* (*Li et al., 2022*). Based on their locations and periodicities along the 96 nm repeat, we identified many conserved MIP structures within the *S. rosetta* flagellar doublet microtubules, including MIPs 1 a, 1b, 2 a, 2b, 2 c, 3 a, 3b, and 6a-d (*Figure 5*, A-F). MIP1a is typically longer than MIP1b in other species (*Song et al., 2020*), but in *S. rosetta*, MIP1a is shorter than MIP1b (*Figure 5*, A-B, D-E). Furthermore, we identified a previously unobserved ~3.5 nm wide filamentous MIP, here named rail-MIP, which runs along the length of the A-tubule near protofilament A13 and seems to connect to MIP 6ab (*Figure 5A and G*, class 2). The electron density of this rail-MIP was reduced in the average of all axonemal repeats (*Figures 3A and 5A*), suggesting its presence on only a subset of repeat units. To further explore this heterogeneity, we performed automated classification analyses (*Heumann et al., 2011*) focused on the rail-MIP by applying a mask around the region of interest and using principle component analyses to sort the results into identifiable features. Indeed, these analyses revealed that the rail-MIP was not ubiquitously present: only 39% of all averaged axonemal repeat units contained the rail-MIP, and its presence was enriched in DMTs 5–7 (*Figure 5G*, class 2 and table), as compared to doublets 1–4 and 8–9, which mostly lacked the rail-MIP (*Figure 5G*, class 1 and table). The rail-MIP distribution varied between tomograms: about half of the tomograms contained prevalent rails concentrated in the microtubules stated above, whereas the other half of flagellar reconstructions contained fewer, scattered rail-MIPs (*Figure 5—figure supplement 1*). This asymmetric distribution does not appear to correlate with any other observed features (such as presence of vane, barbs, microvilli, or IFT particles), but the rail-MIP is present in a cryo-FIB-milled lamella containing the proximal region of the flagellum (*Figure 5—figure supplement 2*), suggesting that its distribution could be related to the location of the tomogram along the length of the flagellum (proximal vs. distal).

Flagellar DMTs of most (wild-type) species described so far by cryo-ET display one ~4 nm long hole per axonemal repeat in the inner junction between protofilaments A1 and B10; the only described exception is the *T. brucei* flagellar DMTs that has an additional (more proximal) inner junction hole per repeat (*Imhof et al., 2019*). We and others have shown that one PACRG-subunit is missing near the N-DRC base-plate from the FAP20-PACRG inner junction filament (*Dymek et al., 2019*; *Ma et al., 2019*; *Nicastro et al., 2011*). In addition to this inner junction-hole near the N-DRC, *S. rosetta* flagella contain two additional inner junction-holes, one near the base of RS1, and one near the base of RS3 (*Figure 5—figure supplement 3B and D*, pink and green arrowheads). Distances between the previously-reported N-DRC-related inner junction-hole and the additional proximal and distal hole are ~32 and ~16 nm, respectively, suggesting that they could represent additional PACRG subunit losses, given the 8 nm periodicity of the FAP20-PACRG repeat (*Dymek et al., 2019*). Notably, the location of the proximal inner junction hole in *S. rosetta* does not correspond to the proximal inner junction-hole in *T. brucei*, which is ~48 nm proximal to the previously reported N-DRC-related inner junction hole (*Imhof et al., 2019*). *S. rosetta* flagellar DMTs also exhibit a (so far unique) ~6.5 nm long gap in protofilament A2 of the A-tubule between RS3 and RS1 from the next axonemal repeat unit, likely due to a missing tubulin dimer (*Figure 5H*, class 2; *Figure 5—figure supplement 3*, B and D, olive arrowheads). Like the heterogeneous rail-MIP, the electron density in the position of the A2-hole was reduced but not completely missing in the average of all axonemal repeats (*Figure 3C*, *Figure 5—figure supplement 3B*), suggesting its presence on only a subset of the repeats. Our classification analyses revealed that the A2-hole is present in ~39% of repeats, including over 50% of repeats from DMTs 1, 5, and 6 and with lower frequencies in the other DMTs (*Figure 5H* table: class 2). Unlike the rail-MIP, the distribution of the A2-hole across tomograms did not cluster, instead

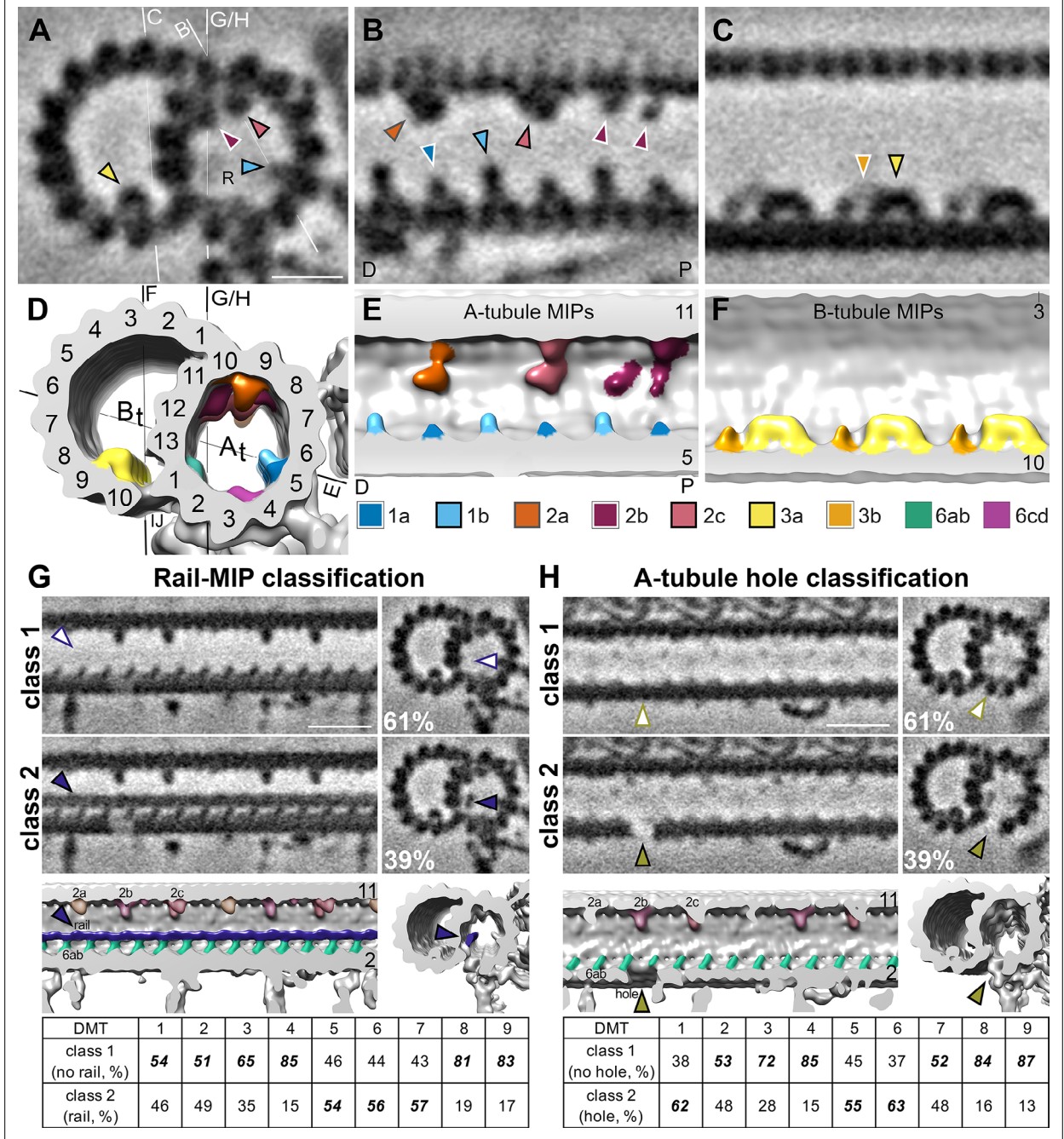

**Figure 5.** *S. rosetta* microtubule doublets contain unique holes and MIPs. (**A–F**). Tomographic slices (**A–C**) and isosurface renderings (**D–F**) of the subtomogram average of *S. rosetta* doublet microtubules shown in cross (**A, D**) and longitudinal (**B, C, E, F**) section at the level of RS1. The white and black lines in (**A**) and (**D**), respectively indicate the viewing positions of the longitudinal slices in (B, C, and E-H). Note: panels (**B and E**) portray the distal (**D**) flagellum to the left, and the proximal (**P**) flagellum to the right. MIPs (and their corresponding arrowheads) are colored as indicated in the legend below panels (E/F). Because the MIPs repeat with a periodicity of 48 nm or less, only a 48 nm long segment of the 96 nm axonemal unit is shown. (**G, H**) Classification analyses focused on the region with the newly identified rail-MIP (**G**) and A-tubule hole (**H**) indicating their presence only in subsets of the axonemal repeats. Class 1 (top rows) lack the rail-MIP or A-tubule hole (empty arrowheads), whereas class 2 (bottom rows) contain the rail-MIP (navy blue arrowhead) or A-tubule hole (olive arrowhead), respectively. Percentages of repeats out of 7584 averaged particles are indicated for each class. The isosurface renderings highlight the position of the rail-MIP (navy blue) between protofilaments A1 and A13, adjacent to MIP 6ab (jade) (**G**), and of the A-tubule hole (olive arrowhead) in protofilament A2 (**H**). The tables show the doublet-specific distribution of the classes. Note: the rail-MIP and A-tubule hole distributions only partially overlap (*Figure 5—figure supplement 1*). *Figure 5—figure supplement 2* indicates the presence of the rail-MIP in the proximal flagellum. *Figure 5—figure supplement 3* shows two additional holes in the *S. rosetta* inner junction. Scale bars: 10 nm (**A**, applies also to **B**, **C**); 20 nm (G, applies to all other images in the panel); 20 nm (H, applies to all other images in the panel).

*Figure 5 continued on next page*

*Figure 5 continued*

The online version of this article includes the following figure supplement(s) for figure 5:

**Figure supplement 1.** Distribution of rail-MIP and A-tubule hole classes by tomogram and doublet microtubule.

**Figure supplement 2.** The rail-MIP is present in the proximal region of the *S. rosetta* flagellum.

**Figure supplement 3.** Additional holes in the doublet inner junction of the *S. rosetta* flagellum.

appearing relatively evenly scattered throughout different tomograms (*Figure 5—figure supplement 1*). Although the DMT-specificity between the rail-MIP and A2-hole somewhat overlapped (presence in DMTs 5 and 6), there was only a mild correlation between these two unique DMT features within repeat units, as evidenced by the hole appearing mildly stronger in class 2 containing the rail-MIP, but still somewhat present in class 1 without the rail-MIP (*Figure 5G–H*, *Figure 5—figure supplement 1*). An additional DMT-specific density corresponds to a protruding structure near the A2 hole, which is present or partially present on DMTs 1, 2, 5, 6, and 7, but only weakly visible on or absent from DMTs 3, 4, 8, and 9 (*Figure 3—figure supplement 2*, navy blue arrowheads).

## The *S. rosetta* central pair complex shows overall conserved features with some reductions

The CPC forms the central core of the axoneme in most flagella and consists of two singlet microtubules (C1 and C2) that are surrounded by a specific set of projections (*Carbajal-González et al., 2013*). In some organisms, the CPC is fixed in its orientation relative to the doublet microtubules, whereas in others, such as *Chlamydomonas*, it twists within the axoneme (*Omoto et al., 1999*). Like other opisthokonts, the *S. rosetta* CPC has a relatively fixed orientation, and the plane that contains both CPC microtubules is roughly parallel to the 5–6 bridge (*Figure 6—figure supplement 1*). This orientation is consistent with *S. rosetta's* flagellum having a planar, sinusoidal waveform (*Dayel et al., 2011*; *Dayel and King, 2014*) with an amplitude (beating direction) that is perpendicular to the CPC and 5–6-bridge planes.

To better resolve the molecular details of the *S. rosetta* CPC, we performed subtomogram averaging of >1300 repeats (32 nm length) that were extracted from 28 cryo-tomograms (selected based on best image signal-to-noise ratio) (*Figure 6*), which yielded an average with 2.5 nm resolution (0.5 FSC criterion) (*Figure 3—figure supplement 1*, *Table 1*). The *S. rosetta* CPC contains all major projections described in other organisms with the previously described longitudinal periodicities of 16 nm (C1a, b; C2a, b, c, d, and e) and 32 nm (C1c, d, e, f) (*Figure 6*, *Figure 6—figure supplement 2*; *Carbajal-González et al., 2013*; *Fu et al., 2019*). In many ways, the *S. rosetta* CPC strongly resembles that of sea urchin sperm flagella (*Carbajal-González et al., 2013*; *Fu et al., 2019*): both lack the C2 MIP, have a small C1e projection, and exhibit prominent connections between the C1a and C2a projections, as compared to the *Chlamydomonas* CPC (*Figure 6*, *Figure 6—figure supplement 2*). One unique feature of the *S. rosetta* CPC, however, is the partial reduction of the C1d protein network, specifically it seems that FAP54 is lacking (*Han et al., 2022*), which exposes a larger area of the C1 microtubule wall (*Figure 6*, *Figure 6—figure supplement 2*).

## The flagellar vane is a bilayer of mesh-like, extracellular filaments

The flagellar vane is a mysterious structure on either side of the proximal area of the choanoflagellate flagellum that has long escaped electron microscopists using traditional methods (*Leadbeater, 2006*). Computer modeling predicts that a vane is necessary to generate fluid motion that would allow bacteria to be phagocytosed by the choanoflagellate microvilli and cell body (*Nielsen et al., 2017*), but the presence of a vane structure itself has only been observed at low resolution on a few species, including *Codosiga botrytis*, *Salipingoeca frequentissima*, *Monosiga brevicolis*, and *Salpingoeca amphoridium* (*Hibberd, 1975*; *Leadbeater, 2006*; *Mah et al., 2014*). In contrast to previous studies in which vane preservation was an issue, we clearly observed vane filaments extending from the flagellar membrane on either side of the flagellum in *S. rosetta*, both in the cryo-FIB lamella of the proximal flagellar region (*Figure 2*, D and F; *Figure 5—figure supplement 2A*), as well as more distally, where the flagellum is embedded in ice thin enough to be directly imaged using cryo-ET (*Figure 2G*, *Figure 7*). The vane originates at the base of the flagellum, and its edges often extend the entire width of (and beyond) our tomograms and lamella, which are ~1.2 μm (tomograms) to ~3 μm

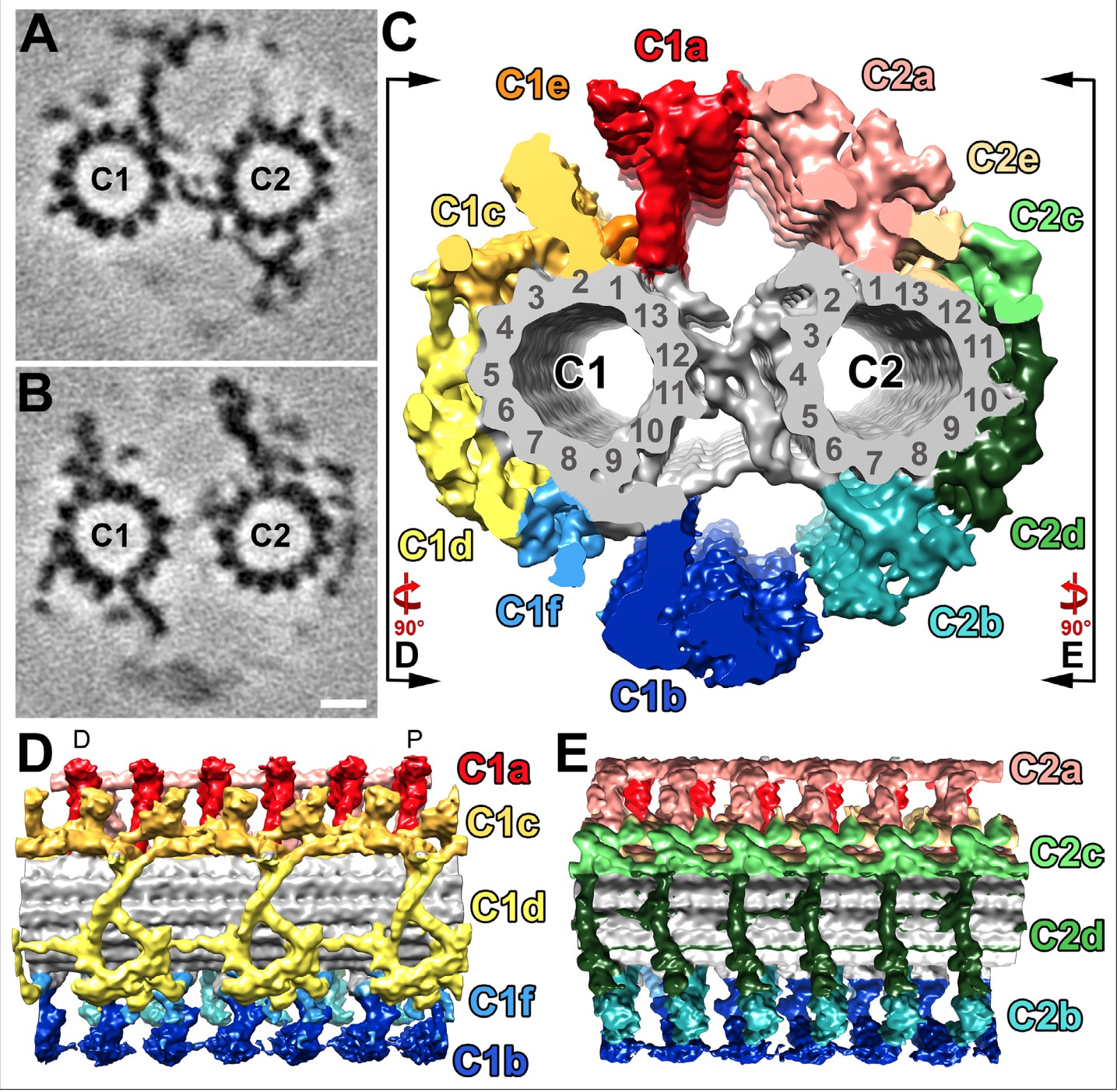

**Figure 6.** Structural features of the *S.rosetta* central pair complex. (**A–B**) Tomographic slices at two different positions of the averaged *S. rosetta* CPC. The slice in (**A**) highlights CPC projections C1a, C2b, and the central bridge, whereas (**B**) highlights C1b-c, C2a, and C2c-e. Averages were generated using 1323 particles from 28 different tomograms (Resolution information in *Figure 3—figure supplement 1* and *Table 1*). (**C**) Isosurface rendering of the averaged central pair complex; projection colors follow (*Carbajal-González et al., 2013*). Black lines and rotation arrows indicate the viewing directions of (**D**) and (**E**). (**D–E**) Isosurface renderings showing longitudinal side-views of the averaged *S. rosetta* CPC. Note: panel (**D**) is oriented with the distal side of the flagellum to the left, and proximal to the right (D and P, respectively). The orientation of the CPC in relation to the 5–6 bridge, vane, and barb structures is shown in *Figure 6—figure supplement 1*. Additional species comparisons are provided in *Figure 6—figure supplement 2*. Scale bar: 10 nm (B, applies also to A).

The online version of this article includes the following figure supplement(s) for figure 6:

**Figure supplement 1.** Relative orientation of CPC to DMTs and extra flagellar features.

*Figure 6 continued on next page*

*Figure 6 continued*

**Figure supplement 2.** Evolutionary comparison of the central pair complex between *Chlamydomonas reinhardtii,* choanoflagellate (*S.rosetta*), and sea urchin (*S. purpuratus*).

(lamella) wide (*Figure 2D–G*, *Figure 5—figure supplement 2A*, *Figure 7*, *Figure 7—figure supplement 1A*). In our data, the plane containing the vane varies in relation to the CPC/sub-5–6 planes, and instead appears to be oriented parallel to the ice layer in which the sample is embedded, likely due to surface tension forces during blotting (*Figure 6—figure supplement 1J-K*). This orientation is consistent with the vane's predicted physiological function in the pumping mechanism of these filter feeders (*Nielsen et al., 2017*), as the vane would be naturally positioned to experience hydrodynamic drag, pushing liquid and prey close to the collar (*Figure 6—figure supplement 1L*).

Our data suggest that the *S. rosetta* flagellar vane is composed of two sheets of thin filaments on either side of the flagellum, which extend from the flagellar membrane for approximately 80 nm before they split and attach to neighboring filaments to form diamond-shaped meshes (*Figure 7A-D*). One or two rows of 'nodes' are visible near the flagellar membrane where the filaments first branch (*Figure 2F*, *Figure 7A*, *Figure 5—figure supplement 2A*). Because our reconstructions are three dimensional, we can view these nodes rotated 90 degrees around the x-axis, which clearly shows that the two sheets of vane filaments and the nodes are separated by ~45 nm (*Figure 7A*, inset). The nodes of the vane filaments are offset from one another so that the vertices of the diamonds from one sheet are centered within the diamonds from the overlaying sheet when viewed from top down (*Figure 7A and C*). Consistent with previous reports, we observed some regions with vane filaments that were highly organized and interconnected, whereas others appeared to have wispy, individual filaments (*Figure 7B-D*). Tomograms often contained areas with wispy hairs and areas with structured vanes, typically on opposite sides of the flagellum (as in *Figure 7A*). Vane filaments were apparent in all but one of the 54 tomograms we analyzed, with most tomograms exhibiting at least partial organized, mesh-like structures (*Figure 7B–D*). We also observed vane filaments within the flagellar pocket at the base of the flagellum (*Figure 7—figure supplement 1A*).

## Previously undescribed barb structures protrude from the *S. rosetta* flagellar membrane

In regions of the flagellar membrane adjacent to where the vane filaments protrude, we also observed previously undescribed barb-like membrane complexes, hereafter denoted as 'barbs', that extend ~50 nm from the extracellular surface of the flagellar membrane (*Figure 7E-I*). To better resolve the molecular details of these barbs, we performed subtomogram averaging and applied fourfold symmetry resulting in 600 averaged particles that yielded an average with 2.5 nm resolution (0.5 FSC criterion) (*Figure 7F–I*; *Table 1*). The barbs consist of a top knob and a central rod with a wider mid-body, from which four arms protrude to connect to the membrane (*Figure 7E-I*). Although the top knob of the barb resembles the size of the nodes of the flagellar vane (~7 nm diameter), barbs were not observed at the base of each vane filament. The number of barbs varied greatly between tomograms (which show ~2 μm flagellar length), ranging from 0 to 22 barbs, but their location was consistently near the base of the vane filaments, with a median distance of 50 nm from the barb structures to the vane plane (*Figure 7I–K*, *Figure 6—figure supplement 1A–I*). Although a lumen is visible throughout parts of the central rod, the cavity/channel does not appear to be continuous, and the base of the rod seems only weakly connected to the membrane, if at all. We also observed several barb-like structures within the flagellar pocket (*Figure 7—figure supplement 1A*). The overall shape of the barbs resembles head-less bacteriophages or bacterial secretion needles, but barb structures were not observed on the surface of 3D reconstructed *E. pacifica* bacterial cells, which were co-cultured with *S. rosetta* as a food source (*Figure 7—figure supplement 1B–C*).

## Discussion

Cilia and flagella are hallmarks of eukaryotic cells, dating back to the LECA (*Cavalier-Smith, 2002*; *Mitchell, 2004*). Flagellar defects disrupt many important cellular functions and cause a variety of diseases in humans, collectively known as ciliopathies (*Reiter and Leroux, 2017*). Though detailed structural information is continually emerging, little is known about high-resolution flagellar ultrastructure

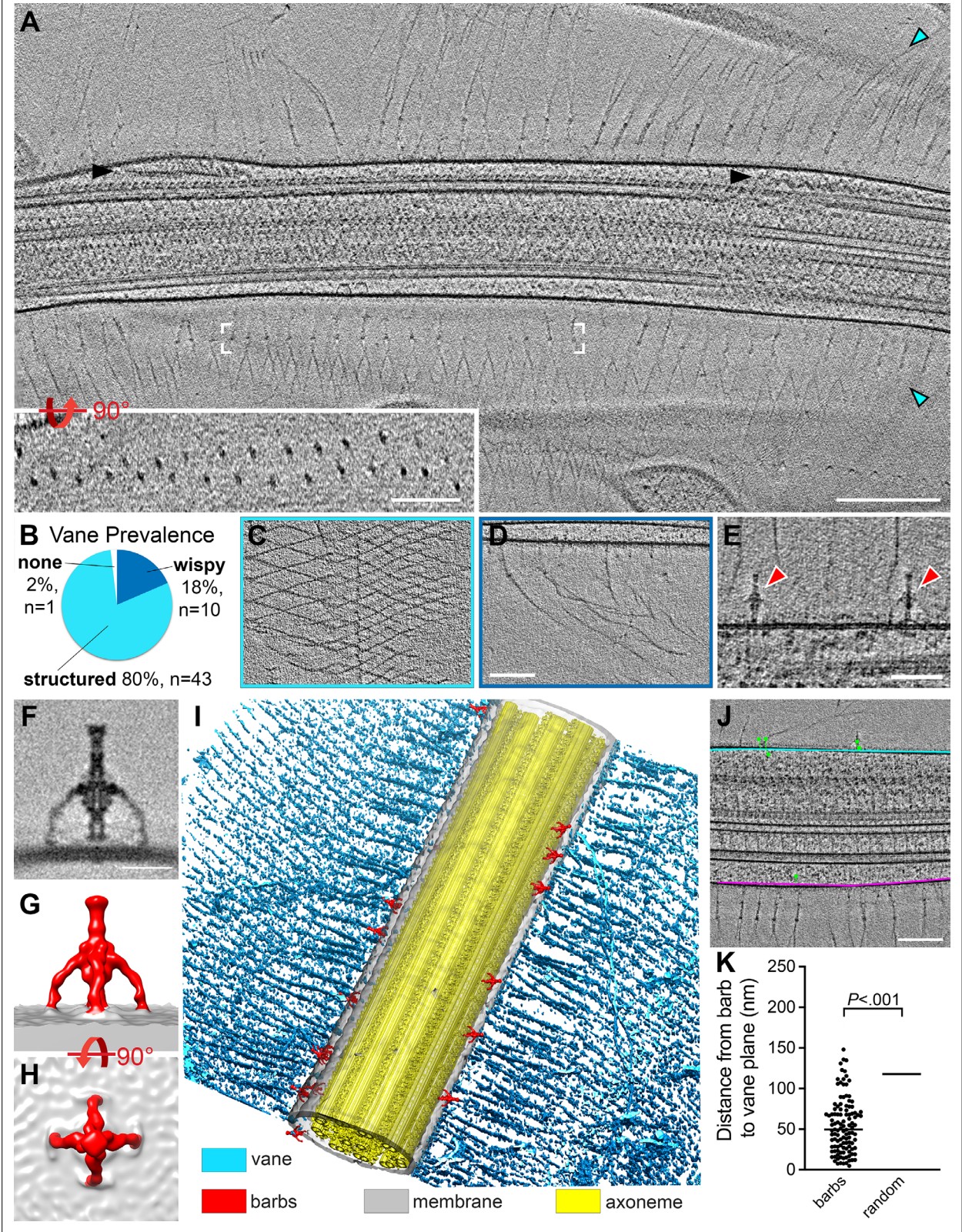

**Figure 7.** *S. rosetta* cells have a flagellar vane and adjacent barb structures. (**A**) Tomographic slice through a representative flagellum showing bilateral vane filaments (cyan arrowheads) extending from the flagellar membrane. Black arrowheads denote IFT trains. White brackets mark the region shown in the rotated inset, which shows that the vane is a bilayer of thin filaments with semi-regular spacing. (**B-D**) Most tomograms contained vane filaments with regular patterning ("structured", light blue color in (**B**), example tomographic slice shown in (**C**)), whereas a smaller proportion contained only

*Figure 7 continued on next page*

*Figure 7 continued*

individual "wispy" hairs (dark blue color in (**B**), example tomographic slice shown in (**D**)). Of the 54 tomograms included in our analyses, only one did not contain vane filaments. (**E**) Tomographic slice showing two barb structures (red arrowheads), approximately 50 nm in height that protrude from the flagellar membrane near the plane of the vane filaments (cyan arrowheads). (**F-H**) Tomographic slice (**F**) and isosurface renderings (**G, H**) show side (F, G) and top (**H**) views of the averaged barb structures (red, 4x symmetrized, 600 particles). (**I**) Compiled isosurface rendering of the *S. rosetta* flagellum, indicating positions of the vane (cyan) and barbs (red) relative to the flagellar membrane (gray); the axoneme is shown in yellow. (**J**) Tomographic slice through a flagellum; the bases of the vane filaments are marked in cyan and pink representing the vane planes; the green dots correspond to the centers of the barb structures in this region (note: the model thickness encompasses the entire flagellum, so most of the barbs themselves are not visible in the tomographic slice, except for the top-right barb). Green lines connect the barb particle to the vane plane with the shortest possible 3D distance (calculated using the mtk function in IMOD). The barb angles cause some to appear inside the membrane, though rotating the model would show that they are indeed protruding externally. (**K**) Quantification of the distances between the barb base to the nearest vane plane for 115 barb particles within 13 tomograms. The black, horizontal lines indicate the median values for the barbs (50 nm, left) compared to a 'random' distribution, which assumes equal likelihood of the barbs being located at any given point around the flagellar circumference (115 nm, right. Individual data points are not shown due to their high number and regular distribution). p=2x10$^{-17}$. *Figure 7—figure supplement 1* shows vane filaments and barbs on the plasma membrane within the flagellar pocket, but not on the surface of *E. pacifica* (bacterial prey). Scale bars: Scale bars: 200 nm (A and inset); 100 nm (D, applies also to C); 50 nm (E); 20 nm (F); 100 nm (J).

The online version of this article includes the following figure supplement(s) for figure 7:

**Figure supplement 1.** Vane filaments and barb-like structures are found in the flagellar pocket of *S.rosetta*, but not on or in the co-cultured bacterial prey cells.

from diverse species. Furthermore, how flagellar ultrastructure may have changed from unicellular to multicellular animals remains unexplored. Within the Opisthokonta clade, high-resolution structures of motile cilia and flagella have been published for several multicellular metazoans, including sea urchins, zebrafish, mouse, pig, horse, and humans (*Leung et al., 2021*; *Lin et al., 2014*; *Nicastro et al., 2006*; *Yamaguchi et al., 2018*; *Zhao et al., 2021*; *Zheng et al., 2021*). However, no unicellular opisthokonts have been studied at similar resolution. Choanoflagellates are the closest living unicellular relatives to metazoans, and choanoflagellate studies have led to countless insights about the origins of multi-cellularity and the evolution of multicellular structures and processes (*King, 2004*). Here, we present high-resolution flagella structures from the choanoflagellate species *S. rosetta*, providing insights into the structural basis for choanoflagellate motility and a foundation to explore the evolution of flagellar ultrastructure between uni- and multi-cellular opisthokonts.

## Flagellar evolution and the last common ancestor between choanoflagellates and metazoans

Though eukaryotic flagella are highly conserved overall, ultrastructural studies have revealed inter-esting dichotomies between unicellular and multicellular specimens. Most unicellular species contain three outer dynein heads (i.e. three dynein heavy chains) per ODA, whereas flagella from multicel-lular species typically contain only two (*Lin et al., 2014*; *Nicastro et al., 2006*; *Pigino et al., 2012*; *Figure 4*). This is consistent with comparative genomic data suggesting that the third outer dynein heavy chain was lost in metazoans and – most likely independently – in excavates (*Kollmar, 2016*). In addition, unicellular organisms like *Tetrahymena* and *Chlamydomonas* have broad RS head structures and connections between all three radial spoke heads, whereas metazoan RS head structures are narrow, RS1 and RS2 are reduced to a pair of thin blades, and RS1 and RS2 are clearly separated from one another (*Figure 4*; *Grossman-Haham et al., 2021*; *Gui et al., 2021*; *Lin et al., 2014*; *Zheng et al., 2021*). Furthermore, the CPC orientation is fixed in metazoans, with stable connections between C1a and C2a and a reduced C1e projection (*Carbajal-González et al., 2013*). Here, we find that the struc-ture and organization of the dyneins, radial spokes, and the CPC observed for metazoan flagella are consistent with those of the unicellular choanoflagellate *S. rosetta*. This suggests that these changes were likely present in the urchoanozoan (the phylogenetic group containing all Choanoflagellates and metazoans), pre-dating the transition to multicellularity.

Why might these ultrastructural changes have occurred? We can speculate that loss of bulkier flagellar structures like the third outer dynein head, broad radial spoke heads, and larger CPC projections may have generated space to accommodate additional molecular components in the common ancestor of choanoflagellates and animals. Although free-swimming unicellular eukaryotes like *Chlamydomonas* also signal through their flagella (sensing light, chemical environmental cues,

and mechanosensory stimuli), cilia and flagella in animals have adapted many additional signaling functions and molecules, including T2R, progesterone receptors, estrogen receptor-ß, interleukin-6 receptor, and Hedgehog (HH) pathway components (*Bloodgood, 2010*; *Mitchell, 2007*; *Sigg et al., 2017*). Many choanoflagellate species spend part or all of their life cycles attached to substrates using carbohydrate-based theca structures or attached to one another in sheets or colonies (*Dayel et al., 2011*; *Leadbeater, 2014*). Could the reduced RS head structures, CPC projections, and dynein motors in *S. rosetta* be related to a shift away from predator avoidance toward increased signaling functions in a more stable and protected environment? For example, the additional force provided by three outer dynein arm motors could help counteract the hydrodynamic effects of multiple cilia and flagella in organisms like *Chlamydomonas* and *Tetrahymena*, whereas the evolutionary selection pressure to retain the third outer dynein head may be lost in organisms with only one flagellum. Another possibility could be that genes encoding the additional outer dynein heavy chain, subunits in the bulkier radial spokes, and/or CPC proteins were linked to genes that were lost for other evolutionarily advantageous reasons. Future studies might examine flagellar structures and beat strength in species from earlier-branching opisthokonts or amoebozoans as well as other sessile filter feeders to expand on these comparisons.

On the other hand, choanoflagellates can exist in free-swimming, single-cell states, and they must find food, avoid predators, and survive harsh aquatic environments like other unicellular organisms. How might they compensate for the decreased flagellar stability that may have resulted from the reduction of flagellar structures (i.e. RS heads, dynein motors, CPC)? We report a unique rail-MIP in *S. rosetta* flagella that runs the length of the A-tubule lumen. MIPs are thought to reinforce structural integrity of doublet microtubules (*Owa et al., 2019*), therefore a combination of the rail-MIP and A2-hole identified here may provide the strength and flexibility necessary to compensate for the loss of bulkier flagellar structures. The rail-MIP is found preferentially in specific doublets with a distribution that resembles the also asymmetric distribution of the beak-MIPs in the B-tubule of DMTs 1,5,6 of *Chlamydomonas* flagella (*Nicastro et al., 2011*). Thus, it is also possible that both the rail-MIP and beak-MIP provide mechanical support to compensate for external forces, such as the extra drag generated by the motion of the flagellar vane (choanoflagellates) or mastigoneme filaments (*Chlamydomonas*) through the aqueous environment, given both structures' wing-like positioning perpendicular to the beating direction of the flagella. Likely, a combination of these factors has enabled choanoflagellates to reduce their dynein and radial spoke structures without losing their cell-propulsion function.

## Enhanced flagellar vane preservation reveals its detailed and unique morphology

The choanoflagellate flagellar vane has remained understudied due to technical challenges with fixation and visualization of this filigree structure. Plunge-freezing and cryo-ET overcome these challenges, allowing us to study the vane in unprecedented detail, both confirming and extending previous observations and interspecies comparisons. The choanoflagellate flagellar vane has only been observed in a few of the >125 known choanoflagellate species (*Hibberd, 1975*; *Leadbeater, 2006*; *Mah et al., 2014*). These reports describe the vane as a bilateral fringe composed of delicate perpendicular fibers of glycocalyx, occasionally with diagonal or longitudinal fibers, which extend approximately two-thirds of the flagellar length (*Hibberd, 1975*; *Leadbeater, 2006*; *Mah et al., 2014*). Our data are consistent with these reports and further resolve two layers of vane filaments on each side of the flagellum, which split and connect with neighboring filaments and are offset to form a mesh-like appearance (*Figure 2D, F and G*, *Figure 7A–D, I*). The wispy hairs we observe appear similar to the 'partially disintegrated' vanes observed in *Monosiga* sp. (*Hibberd, 1975*), suggesting that they are perhaps not disintegrated but rather a common variation on vane structure. We do not know what the wispy vanes represent in relation to the mesh-like vane, but different interpretations are possible: they could be areas of the vane that are broken, areas that are being newly generated or repaired, or perhaps there is some advantage to having meshed vane on one side and wispy vanes on the other.

As has been previously discussed (*Hibberd, 1975*), the structure of the choanoflagellate vane differs significantly from other flagellar appendages, including the hair-like mastigonemes from green algae like *Chlamydomonas* (*Liu et al., 2020*) or the tripartite hairs from golden algae like *Ochromonas* (*Bouck, 1971*). Both the size and arrangement of filaments is distinct, with algal mastigonemes

comprised of two intertwined filaments with an overall diameter of ~10 nm and organized as single or tripartite hairs (*Liu et al., 2020*), vs choanoflagellate filament diameters of ~3.5 nm arranged as wispy hairs or meshed networks. In addition, we do not detect connections between the membrane anchor of the choanoflagellate vane filaments and the axonemal microtubules, in contrast to observations for *Chlamydomonas* and *Euglena* (*Liu et al., 2020*). Consistent with these morphological differences, we also did not detect homologues of the mastigoneme protein MST1 and membrane anchor PKD2 in the *S. rosetta* genome via BLAST search, suggesting that the vane differs from these flagellar appendages. Instead, it was proposed that the morphology of the choanoflagellate vane is similar to the bilateral, wing-like vane of sponge choanocyte flagella, which is anchored in the flagellar membrane without connections to the axonemal microtubules (*Mehl and Reiswig, 1991*). However, sponge choanocyte vanes appear to be narrower, denser, and more massive than choanoflagellate vanes, and they often connect laterally to the collar microvilli (*Brunet and King, 2017*; *Hibberd, 1975*; *Leadbeater, 2006*; *Mah et al., 2014*; *Mehl and Reiswig, 1991*). The choanoflagellate vane has previously been reported to span the width of the collar in *Monosiga brevicolis* (*Mah et al., 2014*), however, the field of view in our tomograms does note capture the ends of the vane to assess any potential connections to the microvilli. Future studies on the ultrastructure of the sponge choanocyte flagellar vane, as well as the composition of both choanocyte and choanoflagellate vane filaments and their lateral connections will further elucidate the extent of their similarity and provide insight as to their evolutionary relationship.

How do these delicate and intricate vane structures form, and what are they made of? Though our data do not directly address these questions, we do observe fibers that originate within the flagellar pocket, suggesting the possibility that the vane could be secreted from the cell membrane before being transported into the flagellar compartment (*Figure 7—figure supplement 1A*). Though not present in our cryo-ET data of fast and slow swimmers, large, fiber-filled vesicles have been observed near the apical part of choanoflagellate cells following division, presumably when the flagellum and vane would be regenerating (*Leadbeater, 2014*). Intriguingly, glycosyltransferases such as those encoded by *jumble* and *couscous* localize to both the basal pole and the flagellar/collar base, and the collar base also stains positively for jacalin, *Lycopersicon esculentum* (tomato) lectin (LEL), and *Solanum tuberosum* (potato) lectin (STL), indicating the presence of carbohydrate chains made of Galß3GalNAc and GlcNAc$_{2-4}$ near the choanoflagellate flagellum (*Wetzel et al., 2018*). We performed an external digest with proteinase K, which failed to remove the flagellar vane (data not shown), supporting the hypothesis that the *S. rosetta* flagellar vane is carbohydrate or glycoprotein-based rather than proteinaceous, though additional study is necessary to further characterize the specific vane component(s).

## Barb structures and their possible functions

In addition to the mysterious composition and function of the flagellar vane (*Leadbeater, 2006*), we have identified previously undescribed barb structures attached to the *S. rosetta* flagellar membrane. At about 50 nm in height and with four 'arms' and a central rod connecting to the flagellar membrane, the barbs' function and composition present yet additional mysteries. Like the vane filaments, we find barb structures in both the flagellar membrane and the plasma membrane of the flagellar pocket (*Figure 7—figure supplement 1A*). Despite of their resemblance to headless bacteriophages or bacterial secretion needles, the barbs' semi-regular distribution in two loose rows around the vane filaments makes it unlikely that the barb structures originate from an external source – for instance the co-cultured *E. pacifica* bacterial prey. Consistently, we do not observe barb structures on the surface or inside of *E. pacifica* cells or floating in the medium (*Figure 7—figure supplement 1B–C*). However, the *S. rosetta* genome has undergone extensive horizontal gene transfer from bacteria and other organisms (*Matriano et al., 2021*), so it is possible that the barbs could have an ancient bacterial origin. Indeed, the barb structures most closely resemble the size, shape, and general distribution of an uncharacterized bacterial complex in *Prosthecobacter debontii*, although the bacterial structures exhibit a fivefold rather than fourfold symmetry (*Dobro et al., 2017*).

The barbs' location alongside the flagellar vane suggests that they could play a role in vane generation or maintenance. Indeed, the top knobs at the barbs' distal ends resemble the size and shape of the nodes within the proximal vane, though the distance to the flagellar membrane in the vane base is roughly twice that of the barb height. Although beyond the scope of this study, finding cellular

contexts in which barbs are increased or decreased, for example during states in which the flagellum and vane are regenerating following cell division or artificially induction could provide additional insight into the barb function(s).

With continually improving cryo-ET workflows and new techniques for genetic modification in choanoflagellates (*Booth and King, 2020*; *Booth et al., 2018*), there has never been a more exciting time for detailed ultrastructural and functional analyses in our closest unicellular relatives. This work raises many important questions, particularly regarding the role of flagellar structural changes and extracellular matrix components in choanoflagellate biology. Combining recent advances in Opisthokonta phylogeny with morphological and ultrastructural traits, we can better predict the nature of the last common ancestor of choanoflagellates and animals. Furthermore, because choanoflagellate flagella more closely resemble human flagella, they may represent an attractive alternative to other protist model systems like *Chlamydomonas* and *Tetrahymena* for the study of human ciliopathies.

# Materials and methods

## Choanoflagellate culture and cryo-preparation

*S. rosetta* co-cultured with *E. pacifica* was obtained from ATCC (PRA-390) and was cultured as previously described (King lab choanoflagellate handbook: https://kinglab.berkeley.edu/resources/, *Levin and King, 2013*). Briefly, cells were maintained in 5% Seawater Complete Media (SWC) diluted with artificial seawater (ASW), both made using Tropic Marin Classic Sea Salt (10134). Cells swimming in the top half of the flask were passed 1:10 to 1:20 every 2–4 days.

Before freezing, cultures were scaled up to 100–400 mL, pelleted at 2000 x g (4 °C, 10–15 min), resuspended in ASW, and starved for 20–24 hr to reduce excess *E. pacifica*. Starved *S. rosetta* cells were then similarly pelleted and resuspended in a small volume of ASW (100 uL – 1 mL), and the concentrated cell sample was mixed 3:1 with 10 nm BSA-coated colloidal gold (*Iancu et al., 2006*) shortly before plunge-freezing. 4 μL of the mixture were pipetted onto a copper EM grid with holey carbon film (R2/2, 200 mesh, Quantifoil Micro Tools GmbH, Q43486) that had been freshly glow-discharged for 30 s at negative 30 mAmp. Samples were back-blotted for 1–3 s with Whatman filter paper (grade 1) to remove excess buffer and immediately plunge frozen into liquid ethane using a homemade plunge freezing device. Vitrified samples were mounted into either Autogrids with notched ring for FIB-milling or regular Autogrids for direct cryo-ET (Thermo Fisher Scientific) and stored in liquid nitrogen until used.

## Cryo-FIB milling

Autogrids with vitrified *S. rosetta* were cryogenically transferred into an Aquilos dual-beam FIB/SEM instrument (Thermo Fisher Scientific) equipped with a cryo-stage that was precooled to –185 °C. To protect the sample and enhance conductivity, layers of platinum were added to the grid surface (sputter-coater: 1 keV and 30 mA for 20 s, gas injection system (GIS): pre-heated to 27 °C and deposited onto the sample for 5 s) (*Schaffer et al., 2017*). An overview image of the grid was generated in SEM mode, and cells suitable for cryo-FIB milling were identified using the Maps software. For milling, the cryo-stage was tilted to a shallow angle of 11–18 degrees between the EM grid and the gallium ion beam. Cryo-FIB milling was performed using a 30 keV gallium ion beam with currents of 30 pA for initial bulk milling and thinning, and 10 pA for final polishing, resulting in ~150-nm-thick, self-supporting lamella. SEM imaging at 3 keV and 25 pA was used to monitor the milling process.

## Cryo-ET imaging

Vitrified samples and cryo-FIB milled lamella were imaged using a 300 keV Titan Krios transmission electron microscope (Thermo Fisher Scientific) equipped with a Bioquantum post-column energy filter (Gatan) used in zero-loss mode with a 20 eV slit width and a Volta Phase Plate with –0.5 μm defocus (*Danev et al., 2014*). The SerialEM microscope control software (*Mastronarde, 2005*) was used to operate the Krios and record dose-symmetric tilt series (*Hagen et al., 2017*) from –60° to +60° tilt with 2° increments. Tilt series images were collected using a K3 Summit direct electron detector (Gatan) at ×26,000 magnification and under low-dose conditions and in counting mode (for each tilt series image: 10 frames, 0.05 s per frame, dose rate of ~28 e/pixel/s, frames were recorded in

super-res mode and then binned by 2, resulting in a pixel size of 3.15 Å). The cumulative electron dose per tilt series was limited to 100 e$^-$/Å$^2$.

## Data processing

Preprocessing and 3D reconstruction were performed using the IMOD software package (*Kremer et al., 1996*). K3 frames were dose-weighted and motion-corrected using Motioncorr2. The tilt series images for whole cell reconstructions were aligned using the 10 nm gold nanoparticles as fiducials. Images for lamella reconstructions were aligned either fiducial-less using patch-tracking in IMOD or using dark features (e.g. from the sputter coat or embedded Gallium) as fiducials. 3D reconstructions were calculated using weighted back-projection. Tomograms were excluded from further analysis if they contained compressed flagella or were damaged by non-vitreous ice. Subtomogram averaging was performed as previously described using the PEET program (*Nicastro et al., 2006*). Initial averages of the barb structures suggested symmetry, thus four-fold symmetry was applied during the final steps of subtomogram averaging. To sort particles (i.e. axonemal repeats) with and without rail-MIP and/or A2 hole, soft-edged masks were applied around those features and unsupervised classification analyses built into the PEET program (*Heumann et al., 2011*) was performed to calculate class averages (*Figure 5G-H*). For clearer views of the radial spoke heads, local alignment refinement was performed focused on each individual RS heads. Isosurface renderings were generated using the UCSF Chimera package software (*Pettersen et al., 2004*). *Figure 7I* contains a compiled isosurface rendering, which uses a single tomogram to indicate the position of the structures so that they are biologically accurate, while substituting the raw data of repetitive structures, such as the barbs and the axonemal and CPC repeat units, with the corresponding higher resolution subtomogram averages using the Chimera software. For the visualization of the vane, the selected raw tomogram was first denoised using Cryo-CARE (*Buchholz et al., 2019*). Briefly, odd and even frames were separately reconstructed and 1200 extracted subvolumes (64 voxels each) were used to train the neuronal network (batch size 16, learning rate 0.0004, 200 epochs, 75 training steps per epoch) with the axoneme masked out to feature the vanes in the cryo-CARE model. The trained network was applied to the full reconstruction to generate a denoised tomogram, which was then used to generate the vane isosurface rendering in Chimera.

Resolution of the 96 nm repeat, CPC, and barb particle were estimated at the base of RS1, C1a projection, and particle center, respectively, using the Fourier shell correlation method with a criterion of 0.5 (*Figure 3—figure supplement 1*, *Table 1*). Tomographic slices (without subtomogram averaging) were denoised for better presentation using either non-linear anisotropic diffusion (*Figure 2E–H* and *Figure 7A and C–E*) or a weighted median filter (smooth filter in IMOD) (*Figure 5—figure supplement 2*, *Figure 6—figure supplement 1A–J*, *Figure 7J*, *Figure 7—figure supplement 1*). To measure the distance from the barb to the vane plane (*Figure 7J-K*), intersections between several vane bases and the flagellar membrane were modeled using IMOD (vane plane), and the 'mtk' command in IMOD was used to find the distance from the base of each barb particle to the nearest intersection with the vane plane.

## Light microscopy

For live-cell imaging, 5–10 µL *S. rosetta* cultures were pipetted directly onto superfrost plus glass slides (Fisherbrand) between two thin streaks of petroleum jelly (applied using a 22-gauge needle with syringe), over which an 18 mm circle glass coverslip (Fisherbrand) was gently suspended to create vertical space for the cells to move freely. Brightfield fast time-lapse series (100 FPS) were acquired on a Nikon Eclipse Lvdia-N microscope equipped with an Andor Zyla 4.2 PLUS sCMOS camera, using a 40x0.75 NA Plan Fluor objective and the Nikon Elements software. Videos were later converted to the.mov format using Fiji. For culture images, *S. rosetta* cells were fixed with 2% glutaraldehyde (Sigma-Aldrich, Germany) for 10 min, and 5–10 µL were transferred to glass slides, covered with a glass coverslip, and sealed with clear nail polish. DIC images were collected on an inverted Nikon Eclipse Ti microscope using a 60x1.4 NA Plan Apochromat oil objective, an Orca-Fusion digital camera (Hamamatsu), and the Nikon Elements software.

## Acknowledgements

The authors acknowledge Drs. Julie Pfeiffer and Arielle Woznica (UT Southwestern Medical Center) for their generous assistance sharing reagents and knowledge of choanoflagellate cell culture, Drs. Jeffrey Woodruff and Maralice Connaci-Sorell (UT Southwestern Medical Center) for sharing equipment, reagents, and advice, and John Heumann and David Mastronarde (University of Colorado, Boulder) for technical advice concerning IMOD. We are also grateful to Dr. Daniel Stoddard, Jose Martinez, Raymond Welch, and Eric Zhang of the Cryo-EM Facility at UT Southwestern Medical Center, as well as current and previous Nicastro lab members, for their ongoing assistance and support. Cryo-EM data were collected at the UT Southwestern Medical Center Cryo-Electron Microscopy Facility, which is supported in part by the CPRIT Core Facility Support Award RP170644. This study was funded by the following grants: National Institutes of Health grants R01GM083122 (to DN) and F32 GM137470 (to JMP), and a Cancer Prevention and Research Institute of Texas (CPRIT) grant RR140082 (to DN). This research was also supported in part by the computational resources provided by the BioHPC super-computing facility located in the Lyda Hill Department of Bioinformatics, UT Southwestern Medical Center. Cryo-ET data for the 96 nm flagellar repeat average, central pair complex average, and barb average have been deposited to the EM Data Bank under accession codes EMD-26204, EMD-26209, and EMD-26210, respectively.

## Additional information

### Funding

| Funder | Grant reference number | Author |
| --- | --- | --- |
| National Institute of General Medical Sciences | R01GM083122 | Daniela Nicastro |
| National Institute of General Medical Sciences | F32 GM137470 | Justine M Pinskey |
| Cancer Prevention and Research Institute of Texas | RR140082 | Daniela Nicastro |

The funders had no role in study design, data collection and interpretation, or the decision to submit the work for publication.

### Author contributions

Justine M Pinskey, Conceptualization, Data curation, Formal analysis, Supervision, Funding acquisition, Validation, Investigation, Visualization, Methodology, Writing – original draft, Project administration, Writing – review and editing; Adhya Lagisetty, Data curation, Formal analysis, Validation, Investigation; Long Gui, Data curation, Formal analysis, Visualization; Nhan Phan, Evan Reetz, Investigation, Methodology; Amirrasoul Tavakoli, Visualization; Gang Fu, Conceptualization, Investigation; Daniela Nicastro, Conceptualization, Supervision, Funding acquisition, Validation, Visualization, Methodology, Project administration, Writing – review and editing

### Author ORCIDs

Justine M Pinskey (ID) http://orcid.org/0000-0001-5656-5519
Daniela Nicastro (ID) http://orcid.org/0000-0002-0122-7173

### Decision letter and Author response

Decision letter https://doi.org/10.7554/eLife.78133.sa1
Author response https://doi.org/10.7554/eLife.78133.sa2

## Additional files

### Supplementary files
• Transparent reporting form

## Data availability

Cryo-ET subtomogram averages have been deposited in the EM Data Bank under accession codes EMD-26204, EMD-26209, and EMD-26210.

The following datasets were generated:

| Author(s) | Year | Dataset title | Dataset URL | Database and Identifier |
|---|---|---|---|---|
| Pinskey JM, Nicastro D | 2022 | Ciliary 96-nm repeat unit from *Salpingoeca rosetta* (choanoflagellate), generated via cryo-electron tomography and subtomogram averaging | https://www.ebi.ac.uk/emdb/EMD-26204 | Electron Microscopy Data Bank, EMD-26204 |
| Pinskey JM, Nicastro D | 2022 | Subtomogram average of central pair complex from *Salpingoeca rosetta* (choanoflagellate) | https://www.ebi.ac.uk/emdb/EMD-26209 | Electron Microscopy Data Bank, EMD-26209 |
| Pinskey JM, Nicastro D | 2022 | Barb-like structure on the external surface of the *Salpingoeca rosetta* (choanoflagellate) ciliary membrane | https://www.ebi.ac.uk/emdb/EMD-26210 | Electron Microscopy Data Bank, EMD-26210 |

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
