## [Editor Report]

This is a thorough, beautiful and compelling study of the flagellar structure of the choanoflagellate *S. rosetta* as reconstituted by cryo-electron tomography (now one of the handful of eukaryotic species whose flagella have been studied in such detail). The findings yield many important new insights of broad interest to the field (such as the similarity of outer dynein arm and radial spoke structure to metazoan cilia, the observation of a flagellar vane in that species, and the presence of mysterious barb structures).

---

## [Decision Letter]

**Decision letter after peer review:**

Thank you for submitting your article "Three-dimensional cilia structures from animals' closest unicellular relatives, the Choanoflagellates" for consideration by *eLife*. Your article has been reviewed by 3 peer reviewers, and the evaluation has been overseen by a Reviewing Editor and Anna Akhmanova as the Senior Editor. The following individuals involved in review of your submission have agreed to reveal their identity: Thibaut Brunet (Reviewer #2); Brooke Morriswood (Reviewer #3).

Essential revisions:

The reviewers agreed this was a beautiful study that does not require any new data. They suggest a number of suggestions for improvements in presentation and analysis listed below.

The reviewers agree that it would be useful if use the term flagellum rather than cilium (please see Reviewer #2 for details).

They also encourage you to address other possible explanations for the lack of the third dynein arm (also see reviewer #2). On this point it's worth noting that the trypanosome 96 nm repeat (Imhof et al., 2019) shows they also lack a 3rd arm, and yet are highly motile and also use their flagella for sensing.

*Reviewer #1 (Recommendations for the authors):*

This manuscript is well written and falls within the scope of *eLife*. I appreciated the inclusion of both maps and validation reports with submission, thank you. Before fully recommending this for publication, I feel some additional analysis and quantification is required to improve the impact of the study.

1. Though not the fault of the authors, I find the use of MIPX nomenclature (e.g. Line 200) a little confusing, given that proteins that make up these MIPs are in some cases becoming clear. Can the authors, where possible, include protein names in parentheses for the MIPs? Would this be possible for the CPC proteins as well?

2. Do differences in *S. rosetta* Rib72 (which I believe is MIP1?) explain why MIP1a is shorter than MIP1b in their cilia (Line 201-203)?

3. The authors mention "some differences" between *S. rosetta* and other cilia, then only mention one (MIP1a/b lengths, see point 2). What are the other differences between *S. rosetta* and other cilia? If these other differences are minor, could the authors clarify that the MIP1a/b is the major difference, other than the novel proteins then described?

4. Could the authors quantify the vane orientation with respect to the central pair in Figure 5 —figure supplement 1? This would better support their conclusions regarding the vane orientation with respect to the CPC (line 294).

5. Could the authors clarify in the text whether the wispy vanes (Line 311) were interspersed with organised vanes on the same cilium, or vanes on different cilia were either wispy or organised? What do these wispy vanes represent?

6. Could the authors analyse the proximity or colocalisation of the barb structures with the vane filament bases to support their assertion that they are found in rows near the vanes themselves (Line 331)?

7. The authors are missing some important references. Could the authors include the following radial spoke structure references in the results (Line 172) and discussion (lines 368-369)? Grossman-Haham et al. 2020 (https://doi.org/10.1038/s41594-020-00519-9), Zheng et al. 2021 (https://doi.org/10.1073/pnas.2021180118), and Gui et al. 2021 (https://doi.org/10.1038/s41594-020-00530-0).

8. The authors' introduction seems to completely omit the high resolution structures of axoneme components, namely the microtubule doublet (Ma et al. 2019), the radial spoke structures (see point 7), the axonemal dynein structures (Walton et al. 2021, Rao et al. 2021) and central pair apparatus structures (Han et al. 2022) that have been recently released. These structures provide important insights into protein placement within the cilia that are complementary to the excellent cryo-ET and bioinformatics studies the authors mention. As such, they should be mentioned in the paragraph talking about ciliary ultrastructure (lines 74-88).

9. The legend for Figure 4 is very long, containing results that should be in the text rather than the legend (e.g. "Overall both features show some doublet-specific distribution…"). Could the authors remove these results from the legend, and work to shorten it?

*Reviewer #2 (Recommendations for the authors):*

As I have no expertise in cryo-EM, I must defer to the other referee(s) to evaluate those specific aspects of the work. As far as I can assess it, the paper comes across as very solid, and I only have minor points (albeit maybe many of them):

Technical points:

– l. 292-294: "The plane containing the vane varies somewhat in relation to the CPC/sub-5-6 planes, but in most tomograms, it appears to be oriented roughly parallel to the CPC within an angle of up to ~30 degrees between the planes" The vane/CPC angle has not been quantified (although it could have been), and the statement would be stronger with a quantification.

Terminology:

– Throughout the paper, the authors refer to the *S. rosetta* flagellum as a "cilium". This word is not commonly used to refer to this structure – although it is homologous to the cilia of ciliates and metazoans. It is of course unfortunate that two different words are in use (with somewhat inconsistent usage between taxonomic groups), but attempts at a single term ("undulipodia" or "cilia" for everything) have regrettably not achieved general use, and the least confusing option remains to refer to the structure under study as a "flagellum" together with an explanation of current usage.

– Similarly, the microvilli as often erroneously referred to as "collars" (the collar is the set of all microvilli) or as "collar tentacles" (which is outdated terminology).

– l. 149, the base of the flagellum is referred to as "the basal pole" – but it is actually the apical pole of the cell

Interpretations

– l. 376-394, the authors speculate that the lack of a 3rd outer dynein arm and the reduced radial spoke of choano/animal flagella (compared to *Chlamydomonas* and Tetrahymena) might reflect a weaker flagellar beat, and a switch (for choanoflagellates) to a sessile lifestyle. This is not convincing, since the authors have imaged free-swimming choanos, not sessile ones, so no adaptation to a sessile lifestyle is expected in this dataset. Moreover, there is (to my knowledge) no direct evidence that the flagellar beat of choanos is actually weaker than the one of *Chlamydomonas*: this is a speculation based on the structure (a reasonable one, but which should be presented as such).

Presentation

– The text only ever refers to figure supplements as a whole, but never to specific panels. This can make it hard to find what the authors are referring to. Two examples: l. 314 should refer to "Figure 6-Figure S1A" and l. 315 to "Figure 6-Figure S1B-C".

– Figure 7 is hard to interpret for non-experts. Only radial spokes are labelled ("RS"), but inner dyneim arms and outer dynein arms are left for the reader to guess. They should be labelled as well (IDA, ODA). Similarly, it is really not obvious to an untrained eye that choanozoans have only 2 ODAs while other eukaryotes have 3. Maybe the distinct dynein arms (per axonemal repeat) should be labelled somehow (different colors?) to clarify that point?

General knowledge points

– In the intro, the authors state that animals "rely on cilia for developmental signaling, mucosal clearance, feeding, and reproduction", by contrast to protists who generally use them for locomotion. This is not quite valid since many metazoans also use cilia for locomotion (ctenophores, gastrotrichs, planarians, placozoans, planktonic larvae of many marine invertebrates (eg annelids, mollusks, echinoderms, hemichordates, sponges, cnidarians)…)

– Also in the intro (l. 74-88), the authors say that previous comparative studies of cilia/flagella have been limited to microscopy and to sequence comparison between a handful of selected proteins. This ignores the several comparative proteomic studies of flagella that have been performed over the years (for example Pazour et al., JCB 2005; Abedin Sigg et al., Dev Cell 2017).

– l. 281-282: "the presence of a vane structure itself has only been shown for a few freshwater species". This is not true, since the list includes Monosiga brevicollis, which is a sea water species.

– l. 386-387: the authors say that choanoflagellate flagella might be intermediate in signalling protein content between other protists and metazoans, since they contain TRP-associated proteins (though they lack Hedgehog and GPCR components). This is not very convincing since *Chlamydomonas* flagella also have quite some TRP channels.

– l. 443: the authors state that the vane filaments "do not connect laterally to collar filaments" in choanos. This is not true in Monosiga, as Mah et al. (2014) reported direct contact between the vane and the base of the microvilli (Figure 3B; though this contact is lost in more distal parts, since the microvilli are not parallel and eventually "fan out"). Similarly, Figure 2G in the present paper shows potential contact points between vane filaments and microvilli (though I'm not sure whether a preparation artifact can be excluded).

– I strongly recommend switching to the word "flagellum" throughout the paper, with an explanation of the contrasted usage of the words "cilium" and "flagellum" in the introduction.

– I suggest adding a quantification of the vane/CPC angle throughout the dataset and plotting it as a histogram or a scatter plot.

– l. 201-203: "However, we also observed some differences, e.g. MIP1a is typically longer than MIP1b in other species (Song et al., 2020), but in *S. rosetta* MIP1a is shorter than MIP1b (Figure 4, A-B, D-E)." Could this be tested/validated by comparison of the relevant protein sequences?

– Regarding the lack of a 3rd outer dynein arm and the reduced radial spokes in choanozoans: if it indeed reflects a weaker flagellar beat (which remains to be tested!), I don't think a switch to a sessile lifestyle or a pivot toward signaling functions are very plausible explanations (for the reasons given above). An alternative I submit to the author's considerations: choanozoans (like all opisthokonts) have a single flagellum per cell, unlike other studied eukaryotes which have 2 (*Chlamydomonas*) or many (ciliates). Could it be that a stronger flagellar beat is needed to counteract the hydrodynamic effect of the neighboring flagellum, but unnecessary when there is only one?

*Reviewer #3 (Recommendations for the authors):*

I have no pressing recommendations. I really enjoyed the paper both in terms of quality and interest. I've provided a few suggestions below that the authors may wish to implement in order to improve the clarity of the manuscript.

1. Figure 1A should be altered I think, as it could more accurately show the phylogenetic relationships between the selected organisms. Both the alveolates and the Archaeplastida belong to the same SAR (stramenopiles, alveolates, rhizaria) supergroup, so they are more closely related to each other. The excavates are (probably) as distantly related to SAR organisms as they are to opisthokonts. To keep the alterations simple, I would recommend moving grouping the alveolates and Archaeplastida together in a single clade on the left-hand side of the panel, and also moving the excavates to the left-hand side. That will better represent the evolutionary distances involved. The unannotated branches (purple, magenta, brown, grey) could simply be removed. Kops et al., 2020 (https://doi.org/10.1016/j.cub.2020.02.021) has an excellent figure that could be used as a template, if the authors wish to make more extensive alterations.

2. I found the figure legends quite difficult to read, because the panel references (A/B/C etc) are jumping around so much. These could perhaps be restructured a bit for improved clarity?

3. Similarly, it's harder for the reader to follow the flow of the data when figure panels are not cited in order. From the Introduction to the first couple of Results sections, the figure/panel citations go; Figure 1, Figure 1D, Figure 1A, Figure 1, Figure 2A-F, Figure 2D. Please ensure that the flow of the manuscript matches the flow of the figures.

4. Figure 6I – I would consider removing the vane from the compiled rendering – there is actually more information available on these filaments in panels A/C/D and the compiled rendering of the vane looks a bit messy and suggests that there is less structure than there actually is.

5. L359-375/Figure 7 – In general, I don't like new information being introduced in the Discussion. Could this section be moved so that it comes between the existing Figures 3 and 4? Given that the authors highlighted the *T. brucei* axoneme in Figure 1A, is there any reason why it was excluded from the analysis here? They would be sampling more eukaryotic biodiversity if it was included, given that Tetrahymena and *Chlamydomonas* both belong to the SAR supergroup.

---

## [Author Response]

Essential revisions:The reviewers agreed this was a beautiful study that does not require any new data. They suggest a number of suggestions for improvements in presentation and analysis listed below.The reviewers agree that it would be useful if use the term flagellum rather than cilium (please see Reviewer #2 for details).

We thank the reviewers for this suggested change in terminology. Theoretically, the term “cilia” includes “eukaryotic flagella”, and we typically use the former term to avoid confusion with “bacterial flagella”, which are of course very different, non-homologous structures. We do, however, understand the historical preference to refer to longer cilia (often) with symmetric waveforms (e.g. sperm flagella, etc.) as “flagella”. Therefore, we have changed all references of choanoflagellate ‘cilia’ to ‘flagella’ in the manuscript, adding the following text at line 55 to address the first instance:

Line 55-56 “Eukaryotic cilia and flagella (terms often used interchangeably) are long, microtubule-based structures that protrude from the cell surface.”

They also encourage you to address other possible explanations for the lack of the third dynein arm (also see reviewer #2). On this point it's worth noting that the trypanosome 96 nm repeat (Imhof et al., 2019) shows they also lack a 3rd arm, and yet are highly motile and also use their flagella for sensing.

We thank the reviewers for raising this important point, and we have revised the text to include alternate explanations based on the reviewers’ suggestions:

Line 779-790: “Could the reduced RS head structures, CPC projections, and dynein motors in *S. rosetta* be related to a shift away from predator avoidance toward increased signaling functions in a more stable and protected environment? For example, the additional force provided by three outer dynein arm motors could help counteract the hydrodynamic effects of multiple cilia and flagella in organisms like *Chlamydomonas* and Tetrahymena, whereas the evolutionary selection pressure to retain the third outer dynein head may be lost in organisms with only one flagellum. Another possibility could be that genes encoding the additional outer dynein heavy chain, subunits in the bulkier radial spokes, and/or CPC proteins were linked to genes that were lost for other evolutionarily advantageous reasons. Future studies might examine flagellar structures and beat strength in species from earlier-branching opisthokonts or amoebozoans as well as other sessile filter feeders to expand on these comparisons.”

Reviewer #1 (Recommendations for the authors):This manuscript is well written and falls within the scope of eLife. I appreciated the inclusion of both maps and validation reports with submission, thank you. Before fully recommending this for publication, I feel some additional analysis and quantification is required to improve the impact of the study.1. Though not the fault of the authors, I find the use of MIPX nomenclature (e.g. Line 200) a little confusing, given that proteins that make up these MIPs are in some cases becoming clear. Can the authors, where possible, include protein names in parentheses for the MIPs? Would this be possible for the CPC proteins as well?

We thank the reviewer for this suggestion. Though we have carefully considered this option, the authors feel that including protein names would be an overinterpretation of our data, and potentially more confusing to readers than it would be helpful. Though it is true that, in some cases, the proteins that make up individual MIPs and CPC proteins have recently been published in other species such as *Chlamydomonas reinhardtii*, how these proteins relate to *S. rosetta* proteins remains unclear (see example below in response to review 1’s point 2). Our level of resolution does not allow us to fit individual protein structures into the MIP densities or even see some of the smaller structures reported in other papers using different methods, thus we hesitate to make unfounded claims about which proteins are present. In addition, the FAP nomenclature is not used in Choanoflagellates, therefore we would have to list both the *Chlamydomonas* protein names as well as the potential *S. rosetta* homologous protein names, which would be lengthy and confusing to most readers. Similar to other organisms (e.g. *Tetrahymena*) with multiple orthologues of ciliary proteins, blast searches of *Chlamydomonas* flagellar proteins against the *S. rosetta* genome often result in multiple significant hits, therefore it would be both difficult and inaccurate to compile a 1:1 list for protein equivalents between species without further experiments (like candidate protein tagging or knockout). Providing a full list of significant blast results for each *Chlamydomonas* protein in *S. rosetta* does not seem generally useful, as readers can blast the individual proteins they are interested in and get the same information themselves. We therefore kept the standard MIP nomenclature based on position rather than speculating about potential protein homologs.

2. Do differences in *S. rosetta* Rib72 (which I believe is MIP1?) explain why MIP1a is shorter than MIP1b in their cilia (Line 201-203)?

In *Chlamydomonas* and *Tetrahymena*, RIB72 or RIB72A/B, respectively, are elongated, multi-domain proteins that span from protofilaments A13-A1-A5 of the A-tubule. In *Tetrahymena*, Rib72 A/B KOs affect the MIP structures historically termed MIP1, 6 and 4, because Rib72 interacts with other MIP proteins such as FAP252, FAP115, and more (Ma et al. 2019; Li et al. 2022). A BLAST search of *Chlamydomonas* RIB72 (GenBank Accession: AAM44303.1) identifies two potential homologues in *S. rosetta*: EFHC1 protein (XP_004991408.1) and uncharacterized protein PTSG_01492 (XP_004997468.1). At 627 and 609 amino acids, respectively, both are slightly shorter but still similar in length to *Chlamydomonas* RIB72, which has 635 amino acids. Unfortunately, without significant additional genetic knockout and imaging experiments, we have no way of knowing if one of these or both homologues are expressed or localized to the Choanoflagellate flagellum. In *Tetrahymena*, both orthologues localize to cilia and alternate along the doublet microtubule; the RIB72A/B domains that are part of MIP1a and b respectively, seem to have asymmetric interactions with an unidentified density bridging between them (Li et al. 2022). This all said, based on our resolution and the uncertainties about homologs and other interacting subunits, we cannot make conclusions about individual protein components of the MIP structures. So while this is a great question and a great point to raise, we do not have a clear answer based on sequence comparison.

3. The authors mention "some differences" between *S. rosetta* and other cilia, then only mention one (MIP1a/b lengths, see point 2). What are the other differences between *S. rosetta* and other cilia? If these other differences are minor, could the authors clarify that the MIP1a/b is the major difference, other than the novel proteins then described?

We thank the reviewer for pointing out this confusing language, which we have now removed. The major differences that we wanted to highlight were as listed – the difference in MIP1a/b lengths, and the rail-MIP, which we have only observed in Choanoflagellates. It is possible that the other MIPs are also different from other species, though the differences listed are the most obvious at this level of resolution.

4. Could the authors quantify the vane orientation with respect to the central pair in Figure 5 —figure supplement 1? This would better support their conclusions regarding the vane orientation with respect to the CPC (line 294).

We thank the reviewer for this excellent suggestion. We have now quantified the vane orientation with respect to the central pair complex, not only for the datasets shown in former Figure 5 —figure supplement 1 (now Figure 6 —figure supplement 1), but for all tomographic reconstructions included in this study. We find that the CPC orientation varies in relation to the ice layer, however, the vane shows a strong orientation preference to be flat within the thin layer of ice that is generated during blotting and plunge freezing of the EM grids (revised Figure 6 —figure supplement 1, panel J-K). We therefore conclude that the orientation of the vane depends mostly on external forces (i.e. in our samples the surface tension of the thin water/ice layer, and for freely swimming cells hydrodynamic forces), rather than a fixed angle/linkage relative to the CPC.

In the case of freely-swimming cells, the CPC plane is perpendicular to the direction of flagellar beating (see revised Figure 6 —figure supplement 1, panel L) – as previously shown e.g., for sea urchin sperm flagella (Lin et al. 2012). The vanes on each side of the flagellum are likely dragged through the surrounding fluid; if the ciliary membrane accommodates some motion of the vane base (as indicated by our data), the hydrodynamic resistance might push the vane base slightly back in the ciliary membrane, but as soon as the flagellar beat direction reverses, so would the direction of the hydrodynamic forces and the vane base would be pushed in the opposite direction. Thus overall, we expect that the vane orientation would fluctuate in free swimming cells around the CPC plane (as indicated in revised Figure 6 —figure supplement 1, panel L cartoon) due to hydrodynamic forces. We have changed the following text to reflect these new data:

Line 471-477: “In our data, the plane containing the vane varies in relation to the CPC/sub-5-6 planes, and instead appears to be oriented parallel to the ice layer in which the sample is embedded, likely due to surface tension forces during blotting (Figure 6—figure supplement 1, J-K). This orientation is consistent with the vane’s predicted physiological function in the pumping mechanism of these filter feeders (Nielsen et al., 2017), as the vane would be naturally positioned to experience hydrodynamic drag, pushing liquid and prey close to the collar (Figure 6—figure supplement 1, L).”

5. Could the authors clarify in the text whether the wispy vanes (Line 311) were interspersed with organised vanes on the same cilium, or vanes on different cilia were either wispy or organised? What do these wispy vanes represent?

We thank the reviewer for this suggestion, and we now better describe the wispy vanes in the following text:

Line 615-621: “Consistent with previous reports, we observed some regions with vane filaments that were highly organized and interconnected, whereas others appeared to have wispy, individual filaments (Figure 7, B-D). Tomograms often contained areas with wispy hairs and areas with structured vanes, typically on opposite sides of the flagellum (as in Figure 7A). Vane filaments were apparent in all but one of the 54 tomograms we analyzed, with most tomograms exhibiting at least partial organized, mesh-like structures (Figure 7 B-D).”

Line 847-859: “We do not know what the wispy vanes represent in relation to the mesh-like vane, but different interpretations are possible: they could be areas of the vane that are broken, areas that are being newly generated or repaired, or perhaps there is some advantage to having meshed vane on one side and wispy vanes on the other.”

6. Could the authors analyse the proximity or colocalisation of the barb structures with the vane filament bases to support their assertion that they are found in rows near the vanes themselves (Line 331)?

We thank the reviewer for this helpful suggestion. We have now incorporated a quantification of the distance from the barb particles to the plane of the vane filament bases (Figure 7J-K). Though the examples with the most barbs (e.g. Figure 7I) appear to have loosely organized rows of barbs, it is difficult to make that assertion for most of the tomograms with barbs, so we have removed the language around being in “rows”. However, our data do support that the barbs are near where the vane filaments are anchored in the ciliary membrane with a median distance of 50 nm. The relevant text now reads:

Lines 632-689: “Although the top knob of the barb resembles the size of the nodes of the flagellar vane (~7 nm diameter), barbs were not observed at the base of each vane filament. The number of barbs varied greatly between tomograms (which show ~2 µm flagellar length), ranging from 0-22 barbs, but their location was consistently near the base of the vane filaments, with a median distance of 50 nm from the barb structures to the vane plane (Figure 7I-K, Figure 6—figure supplement 1A-I).”

7. The authors are missing some important references. Could the authors include the following radial spoke structure references in the results (Line 172) and discussion (lines 368-369)? Grossman-Haham et al. 2020 (https://doi.org/10.1038/s41594-020-00519-9), Zheng et al. 2021 (https://doi.org/10.1073/pnas.2021180118), and Gui et al. 2021 (https://doi.org/10.1038/s41594-020-00530-0).

We thank the reviewer for identifying these missing references and we have inserted them where indicated (now corresponding to lines 274-276 and 711-712).

8. The authors' introduction seems to completely omit the high resolution structures of axoneme components, namely the microtubule doublet (Ma et al. 2019), the radial spoke structures (see point 7), the axonemal dynein structures (Walton et al. 2021, Rao et al. 2021) and central pair apparatus structures (Han et al. 2022) that have been recently released. These structures provide important insights into protein placement within the cilia that are complementary to the excellent cryo-ET and bioinformatics studies the authors mention. As such, they should be mentioned in the paragraph talking about ciliary ultrastructure (lines 74-88).

We thank the reviewer for identifying these missing references and we have inserted them where indicated (now corresponding to lines 93-94).

9. The legend for Figure 4 is very long, containing results that should be in the text rather than the legend (e.g. "Overall both features show some doublet-specific distribution…"). Could the authors remove these results from the legend, and work to shorten it?

We thank the reviewer for this suggestion. We removed the indicated section and identified additional ways to shorten the figure legend for revised Figure 5 (formerly Figure 4). Edits can be found on page 35 of the manuscript text.

Reviewer #2 (Recommendations for the authors):As I have no expertise in cryo-EM, I must defer to the other referee(s) to evaluate those specific aspects of the work. As far as I can assess it, the paper comes across as very solid, and I only have minor points (albeit maybe many of them):Technical points:– l. 292-294: "The plane containing the vane varies somewhat in relation to the CPC/sub-5-6 planes, but in most tomograms, it appears to be oriented roughly parallel to the CPC within an angle of up to ~30 degrees between the planes" The vane/CPC angle has not been quantified (although it could have been), and the statement would be stronger with a quantification.

We thank the reviewer for this excellent suggestion. We have now quantified the vane orientation with respect to the central pair complex, not only for the datasets shown in former Figure 5 —figure supplement 1 (now Figure 6 —figure supplement 1), but for all tomographic reconstructions included in this study. We find that the CPC orientation varies in relation to the ice layer, however, the vane shows a strong orientation preference to be flat within the thin layer of ice that is generated during blotting and plunge freezing of the EM grids (revised Figure 6 —figure supplement 1, panel J-K). We therefore conclude that the orientation of the vane depends mostly on external forces (i.e. in our samples the surface tension of the thin water/ice layer, and for freely swimming cells hydrodynamic forces), rather than a fixed angle/linkage relative to the CPC.

In the case of freely-swimming cells, the CPC plane is perpendicular to the direction of flagellar beating (see revised Figure 6 —figure supplement 1, panel L) – as previously shown e.g., for sea urchin sperm flagella (Lin et al. 2012). The vanes on each side of the flagellum are likely dragged through the surrounding fluid; if the ciliary membrane accommodates some motion of the vane base (as indicated by our data), the hydrodynamic resistance might push the vane base slightly back in the ciliary membrane, but as soon as the flagellar beat direction reverses, so would the direction of the hydrodynamic forces and the vane base would be pushed in the opposite direction. Thus overall, we expect that the vane orientation would fluctuate in free swimming cells around the CPC plane (as indicated in revised Figure 6 —figure supplement 1, panel L cartoon) due to hydrodynamic forces. We have changed the following text to reflect these new data:

Line 471-477: ”In our data, the plane containing the vane varies in relation to the CPC/sub-5-6 planes, and instead appears to be oriented parallel to the ice layer in which the sample is embedded, likely due to surface tension forces during blotting (Figure 6—figure supplement 1, J-K). This orientation is consistent with the vane’s predicted physiological function in the pumping mechanism of these filter feeders (Nielsen et al., 2017), as the vane would be naturally positioned to experience hydrodynamic drag, pushing liquid and prey close to the collar (Figure 6—figure supplement 1, L).”

Terminology:– Throughout the paper, the authors refer to the *S. rosetta* flagellum as a "cilium". This word is not commonly used to refer to this structure – although it is homologous to the cilia of ciliates and metazoans. It is of course unfortunate that two different words are in use (with somewhat inconsistent usage between taxonomic groups), but attempts at a single term ("undulipodia" or "cilia" for everything) have regrettably not achieved general use, and the least confusing option remains to refer to the structure under study as a "flagellum" together with an explanation of current usage.

We appreciate this suggestion and have changed the term “cilium” to “flagellum” where applicable throughout the manuscript to adhere to common (historical) usage and avoid confusion. We also added the following text to clarify usage:

Line 55-56 “Eukaryotic cilia and flagella (terms often used interchangeably) are long, microtubulebased structures that protrude from the cell surface.”

– Similarly, the microvilli as often erroneously referred to as "collars" (the collar is the set of all microvilli) or as "collar tentacles" (which is outdated terminology).

We thank the reviewer for raising these terminological issues. We now refer to the microvilli appropriately in all instances.

– l. 149, the base of the flagellum is referred to as "the basal pole" – but it is actually the apical pole of the cell

We thank the reviewer for identifying this issue. We have changed basal to apical accordingly.

Line 226-227: “We observe multiple microvilli bases and many vesicles distributed throughout the apical end of the cell (Figure 2E).”

Interpretations– l. 376-394, the authors speculate that the lack of a 3rd outer dynein arm and the reduced radial spoke of choano/animal flagella (compared to Chlamydomonas and Tetrahymena) might reflect a weaker flagellar beat, and a switch (for choanoflagellates) to a sessile lifestyle. This is not convincing, since the authors have imaged free-swimming choanos, not sessile ones, so no adaptation to a sessile lifestyle is expected in this dataset. Moreover, there is (to my knowledge) no direct evidence that the flagellar beat of choanos is actually weaker than the one of Chlamydomonas: this is a speculation based on the structure (a reasonable one, but which should be presented as such).

The reviewer is correct to point out that this statement is speculatory, and that we do not address beat strength directly (nor is this information available in the literature to our knowledge). We have therefore revised the text to encompass this and Reviewer 2’s last point. It now reads:

Line 779-790: “Could the reduced RS head structures, CPC projections, and dynein motors in *S. rosetta* be related to a shift away from predator avoidance toward increased signaling functions in a more stable and protected environment? For example, the additional force provided by three outer dynein arm motors could help counteract the hydrodynamic effects of multiple cilia and flagella in organisms like *Chlamydomonas* and Tetrahymena, whereas the evolutionary selection pressure to retain the third outer dynein head may be lost in organisms with only one flagellum. Another possibility could be that genes encoding the additional outer dynein heavy chain, subunits in the bulkier radial spokes, and/or CPC proteins were linked to genes that were lost for other evolutionarily advantageous reasons. Future studies might examine flagellar structures and beat strength in species from earlier-branching opisthokonts or amoebozoans as well as other sessile filter feeders to expand on these comparisons.”

Presentation– The text only ever refers to figure supplements as a whole, but never to specific panels. This can make it hard to find what the authors are referring to. Two examples: l. 314 should refer to "Figure 6-Figure S1A" and l. 315 to "Figure 6-Figure S1B-C".

We thank the reviewer for pointing this out and we have updated the text to include references to specific figure panels in the locations indicated as well as throughout the manuscript where appropriate.

– Figure 7 is hard to interpret for non-experts. Only radial spokes are labelled ("RS"), but inner dyneim arms and outer dynein arms are left for the reader to guess. They should be labelled as well (IDA, ODA). Similarly, it is really not obvious to an untrained eye that choanozoans have only 2 ODAs while other eukaryotes have 3. Maybe the distinct dynein arms (per axonemal repeat) should be labelled somehow (different colors?) to clarify that point?

We thank the reviewer for raising these excellent critiques. We now include labels in former Figure 7 (revised Figure 4) panel A (IDA and ODA), we have colored the 2 or 3 different ODA dynein heads with slightly different colors, and we indicate the ODA heads with 2 or 3 arrowheads to help readers recognize these structures.

General knowledge points– In the intro, the authors state that animals "rely on cilia for developmental signaling, mucosal clearance, feeding, and reproduction", by contrast to protists who generally use them for locomotion. This is not quite valid since many metazoans also use cilia for locomotion (ctenophores, gastrotrichs, planarians, placozoans, planktonic larvae of many marine invertebrates (eg annelids, mollusks, echinoderms, hemichordates, sponges, cnidarians)…)

We have edited the text to address this valid criticism:

Line 59-63: “The vast majority of eukaryotic life consists of unicellular organisms with flagella, which perform a variety of functions necessary for their survival, e.g., aid in motility, feeding, avoiding predators, and sensing the environment (Burki, 2014; Mitchell, 2007). Multicellular eukaryotes, including animals (metazoans), also rely on cilia and flagella for locomotion, developmental signaling, mucosal clearance, feeding, and reproduction.”

– Also in the intro (l. 74-88), the authors say that previous comparative studies of cilia/flagella have been limited to microscopy and to sequence comparison between a handful of selected proteins. This ignores the several comparative proteomic studies of flagella that have been performed over the years (for example Pazour et al., JCB 2005; Abedin Sigg et al., Dev Cell 2017).

The reviewer is correct to point out that these studies should also be included, and we have updated the text accordingly:

Line 103-114: “Our understanding of flagellar ultrastructure and evolution is continually expanding through application of new technologies. Historically, much of our knowledge of flagellar architecture from diverse species has been based on conventional light and electron microscopy studies, which are inherently limited by detection limits and preservation artifacts. Protein sequence comparisons have also yielded important insights, particularly into dynein evolution in eukaryotic flagella (Kollmar, 2016), although this required manual annotation of thousands of genes from hundreds of species, not particularly sustainable for examining hundreds of flagellar proteins. Similarly, comparative proteomic studies have also largely contributed to our understanding of flagella composition and evolution (Pazour et al., 2005; Sigg et al., 2017), though both sequence comparisons and proteomics are limited in their ability to predict protein localization and interactions. As a result, our knowledge of detailed flagellar morphology, function, and evolution has remained restricted..”

– l. 281-282: "the presence of a vane structure itself has only been shown for a few freshwater species". This is not true, since the list includes Monosiga brevicollis, which is a sea water species.

We thank the reviewer for correcting our error (we had used AlgaeBase to check *Monosiga brevicollis’* environment, which incorrectly lists it as a freshwater species: https://www.algaebase.org/search/species/detail/?species_id=149216). The corrected text now reads:

Line 443-466: “Computer modeling predicts that a vane is necessary to generate fluid motion that would allow bacteria to be phagocytosed by the choanoflagellate microvilli and cell body (Nielsen et al., 2017), but the presence of a vane structure itself has only been observed at low resolution on a few species, including Codosiga botrytis, Salipingoeca frequentissima, Monosiga brevicolis, and Salpingoeca amphoridium (Hibberd, 1975; Leadbeater, 2006; Mah et al., 2014). In contrast to previous studies in which vane preservation was an issue, we clearly observed vane filaments extending from the flagellar membrane on either side of the flagellum in *S. rosetta*, …”

– l. 386-387: the authors say that choanoflagellate flagella might be intermediate in signalling protein content between other protists and metazoans, since they contain TRP-associated proteins (though they lack Hedgehog and GPCR components). This is not very convincing since Chlamydomonas flagella also have quite some TRP channels.

We thank the reviewer for pointing this out. We have revised the text as follows:

Line 749-757: “Why might these ultrastructural changes have occurred? We can speculate that loss of bulkier flagellar structures like the third outer dynein head, broad radial spoke heads, and larger CPC projections may have generated space to accommodate additional molecular components in the common ancestor of choanoflagellates and animals. Although free-swimming unicellular eukaryotes, like *Chlamydomonas*, also signal through their flagella (sensing light, chemical environmental cues, and mechanosensory stimuli), cilia and flagella in animals have adapted many additional signaling functions and molecules, including T2R, progesterone receptors, estrogen receptor-ß, interleukin-6 receptor, and Hedgehog (HH) pathway components (Bloodgood, 2010; Mitchell, 2007; Sigg et al., 2017).”

– l. 443: the authors state that the vane filaments "do not connect laterally to collar filaments" in choanos. This is not true in Monosiga, as Mah et al. (2014) reported direct contact between the vane and the base of the microvilli (Figure 3B; though this contact is lost in more distal parts, since the microvilli are not parallel and eventually "fan out"). Similarly, Figure 2G in the present paper shows potential contact points between vane filaments and microvilli (though I'm not sure whether a preparation artifact can be excluded).

We thank the reviewer for raising this point. We have reviewed our tomograms and we do not see any examples of vane filaments directly connecting to collar microvilli. That said, the collar microvilli of unperturbed cells would form a cone/ring around the flagellum. Our tomograms are centered on the flagella, and as mentioned in lines 451-454, the vane extends beyond the boundaries of the tomograms (>3 µm). Presumably, any lateral connections from vane to microvilli would occur at the outermost ends of the vane filaments, which are not captured within our datasets. The widest example we have is the image in Figure 2D, but the vane extends even beyond the edge of that image (and 3D data is only available in the areas marked with green rectangles where tilt series were recorded). In addition, the collar microvilli that we observe in our data are typically not too far above or below the vane, often irregularly spaced and positioned at different angles, indicating that the microvillar organization maybe disrupted (i.e. the cone arrangement flattened) during blotting and freezing. Therefore, delicate connections between vane and collar could be broken as well during sample preparation. As a result, we have removed the statement from line 773 that vane filaments do not connect laterally to collar microvilli and revised the text as follows:

Line 861-873: “However, sponge choanocyte vanes appear to be narrower, denser, and more massive than choanoflagellate vanes, and they often connect laterally to the collar microvilli (Brunet and King, 2017; Hibberd, 1975; Leadbeater, 2006; Mah et al., 2014; Mehl and Reiswig, 1991). The choanoflagellate vane has previously been reported to span the width of the collar in Monosiga brevicolis (Mah et al., 2014), however, the field of view in our tomograms does note capture the ends of the vane to assess any potential connections to the microvilli. Future studies on the ultrastructure of the sponge choanocyte flagellar vane, as well as the composition of both choanocyte and choanoflagellate vane filaments and their lateral connections will further elucidate the extent of their similarity and provide insight as to their evolutionary relationship.”

– I strongly recommend switching to the word "flagellum" throughout the paper, with an explanation of the contrasted usage of the words "cilium" and "flagellum" in the introduction.

We thank the reviewers for this suggested change in terminology. Theoretically, the term “cilia” includes “eukaryotic flagella”, and we typically use the former term to avoid confusion with “bacterial flagella”, which are of course very different, non-homologous structures. We do, however, understand the historical preference to refer to longer cilia (often) with symmetric waveforms (e.g. sperm flagella, etc.) as “flagella”. Therefore, we have changed all references of choanoflagellate ‘cilia’ to ‘flagella’ in the manuscript, adding the following text at line 55 to address the first instance:

Line 55-56 “Eukaryotic cilia and flagella (terms often used interchangeably) are long, microtubulebased structures that protrude from the cell surface.”

– I suggest adding a quantification of the vane/CPC angle throughout the dataset and plotting it as a histogram or a scatter plot.

We thank the reviewer for this excellent suggestion. We have now quantified the vane orientation with respect to the central pair complex, not only for the datasets shown in former Figure 5 —figure supplement 1 (now Figure 6 —figure supplement 1), but for all tomographic reconstructions included in this study. We find that the CPC orientation varies in relation to the ice layer, however, the vane shows a strong orientation preference to be flat within the thin layer of ice that is generated during blotting and plunge freezing of the EM grids (revised Figure 6 —figure supplement 1, panel J-K). We therefore conclude that the orientation of the vane depends mostly on external forces (i.e. in our samples the surface tension of the thin water/ice layer, and for freely swimming cells hydrodynamic forces), rather than a fixed angle/linkage relative to the CPC.

In the case of freely-swimming cells, the CPC plane is perpendicular to the direction of flagellar beating (see revised Figure 6 —figure supplement 1, panel L) – as previously shown e.g., for sea urchin sperm flagella (Lin et al. 2012). The vanes on each side of the flagellum are likely dragged through the surrounding fluid; if the ciliary membrane accommodates some motion of the vane base (as indicated by our data), the hydrodynamic resistance might push the vane base slightly back in the ciliary membrane, but as soon as the flagellar beat direction reverses, so would the direction of the hydrodynamic forces and the vane base would be pushed in the opposite direction. Thus overall, we expect that the vane orientation would fluctuate in free swimming cells around the CPC plane (as indicated in revised Figure 6 —figure supplement 1, panel L cartoon) due to hydrodynamic forces. We have changed the following text to reflect these new data:

Line 471-477: ”In our data, the plane containing the vane varies in relation to the CPC/sub-5-6 planes, and instead appears to be oriented parallel to the ice layer in which the sample is embedded, likely due to surface tension forces during blotting (Figure 6—figure supplement 1, J-K). This orientation is consistent with the vane’s predicted physiological function in the pumping mechanism of these filter feeders (Nielsen et al., 2017), as the vane would be naturally positioned to experience hydrodynamic drag, pushing liquid and prey close to the collar (Figure 6—figure supplement 1, L).”

– l. 201-203: "However, we also observed some differences, e.g. MIP1a is typically longer than MIP1b in other species (Song et al., 2020), but in *S. rosetta* MIP1a is shorter than MIP1b (Figure 4, A-B, D-E)." Could this be tested/validated by comparison of the relevant protein sequences?

In *Chlamydomonas* and *Tetrahymena*, RIB72 or RIB72A/B, respectively, are elongated, multi-domain proteins that span from protofilaments A13-A1-A5 of the A-tubule. In *Tetrahymena*, Rib72 A/B KOs affect the MIP structures historically termed MIP1, 6 and 4, because Rib72 interacts with other MIP proteins such as FAP252, FAP115, and more (Ma et al. 2019; Li et al. 2022). A BLAST search of *Chlamydomonas* RIB72 (GenBank Accession: AAM44303.1) identifies two potential homologues in *S. rosetta*: EFHC1 protein (XP_004991408.1) and uncharacterized protein PTSG_01492 (XP_004997468.1). At 627 and 609 amino acids, respectively, both are slightly shorter but still similar in length to *Chlamydomonas* RIB72, which has 635 amino acids. Unfortunately, without significant additional genetic knockout and imaging experiments, we have no way of knowing if one of these or both homologues are expressed or localized to the Choanoflagellate flagellum. In *Tetrahymena*, both orthologues localize to cilia and alternate along the doublet microtubule; the RIB72A/B domains that are part of MIP1a and b respectively, seem to have asymmetric interactions with an unidentified density bridging between them (Li et al. 2022). This all said, based on our resolution and the uncertainties about homologs and other interacting subunits, we cannot make conclusions about individual protein components of the MIP structures. So while this is a great question and a great point to raise, we do not have a clear answer based on sequence comparison.

– Regarding the lack of a 3rd outer dynein arm and the reduced radial spokes in choanozoans: if it indeed reflects a weaker flagellar beat (which remains to be tested!), I don't think a switch to a sessile lifestyle or a pivot toward signaling functions are very plausible explanations (for the reasons given above). An alternative I submit to the author's considerations: choanozoans (like all opisthokonts) have a single flagellum per cell, unlike other studied eukaryotes which have 2 (Chlamydomonas) or many (ciliates). Could it be that a stronger flagellar beat is needed to counteract the hydrodynamic effect of the neighboring flagellum, but unnecessary when there is only one?

We thank the reviewer for sharing this idea, and we have added this point as well as another possibility as follows:

Line 779-790: “Could the reduced RS head structures, CPC projections, and dynein motors in *S. rosetta* be related to a shift away from predator avoidance toward increased signaling functions in a more stable and protected environment? For example, the additional force provided by three outer dynein arm motors could help counteract the hydrodynamic effects of multiple cilia and flagella in organisms like *Chlamydomonas* and Tetrahymena, whereas the evolutionary selection pressure to retain the third outer dynein head may be lost in organisms with only one flagellum. Another possibility could be that genes encoding the additional outer dynein heavy chain, subunits in the bulkier radial spokes, and/or CPC proteins were linked to genes that were lost for other evolutionarily advantageous reasons. Future studies might examine flagellar structures and beat strength in species from earlier-branching opisthokonts or amoebozoans as well as other sessile filter feeders to expand on these comparisons.”

Reviewer #3 (Recommendations for the authors):I have no pressing recommendations. I really enjoyed the paper both in terms of quality and interest. I've provided a few suggestions below that the authors may wish to implement in order to improve the clarity of the manuscript.

We thank the reviewer for their kind words and for the clarifying suggestions provided below.

1. Figure 1A should be altered I think, as it could more accurately show the phylogenetic relationships between the selected organisms. Both the alveolates and the Archaeplastida belong to the same SAR (stramenopiles, alveolates, rhizaria) supergroup, so they are more closely related to each other. The excavates are (probably) as distantly related to SAR organisms as they are to opisthokonts. To keep the alterations simple, I would recommend moving grouping the alveolates and Archaeplastida together in a single clade on the left-hand side of the panel, and also moving the excavates to the left-hand side. That will better represent the evolutionary distances involved. The unannotated branches (purple, magenta, brown, grey) could simply be removed. Kops et al., 2020 (https://doi.org/10.1016/j.cub.2020.02.021) has an excellent figure that could be used as a template, if the authors wish to make more extensive alterations.

We thank the reviewer for their helpful suggestion. The reviewer is correct to point out that SAR and Archaeplastida are more closely related than other groups. We have revised the diagram in Figure 1 to reflect this.

2. I found the figure legends quite difficult to read, because the panel references (A/B/C etc) are jumping around so much. These could perhaps be restructured a bit for improved clarity?

We thank the reviewer for this helpful comment. We have restructured the figure legends to be more straightforward and un-bolded additional references to each figure panel to make it easier for readers to find the main panel explanations they may be looking for.

3. Similarly, it's harder for the reader to follow the flow of the data when figure panels are not cited in order. From the Introduction to the first couple of Results sections, the figure/panel citations go; Figure 1, Figure 1D, Figure 1A, Figure 1, Figure 2A-F, Figure 2D. Please ensure that the flow of the manuscript matches the flow of the figures.

We thank the reviewer for pointing this out. We have revised the text accordingly, and now the figure/panel citations go: Figure 1A, Figure 1B-D + Figure 1 supplements 1 and 2, Figure 1 supplement 2, Figure 2A-C, Figure 2D-F, and so on. Because the structures we discuss in the text are often present in multiple figures, we do refer to each figure in which the structures are visible, which we acknowledge could be confusing but unfortunately is unavoidable due to the nature of our data. Hopefully the revisions still improve the reader’s experience.

4. Figure 6I – I would consider removing the vane from the compiled rendering – there is actually more information available on these filaments in panels A/C/D and the compiled rendering of the vane looks a bit messy and suggests that there is less structure than there actually is.

We thank the reviewer for highlighting this area of improvement. We have tested additional denoising methods for the raw tomogram to enhance the vane contrast (nonlinear anisotropic diffusion, SIRT reconstruction, CryoCARE – deep learning content-aware denoising), and we have applied more selective masks to better capture the vane structure (Revised figure 7I). Altogether, we think these changes improve the look of the vane within this figure, and we think it is valuable to include it to provide a more holistic understanding of other structures (*e.g*. CPC, barbs) in relation to the vane.

5. L359-375/Figure 7 – In general, I don't like new information being introduced in the Discussion. Could this section be moved so that it comes between the existing Figures 3 and 4? Given that the authors highlighted the *T. brucei* axoneme in Figure 1A, is there any reason why it was excluded from the analysis here? They would be sampling more eukaryotic biodiversity if it was included, given that Tetrahymena and Chlamydomonas both belong to the SAR supergroup.

We thank the reviewer for these suggestions, and we have moved the previous figure 7 (now figure 4) between figures 3 and the previous figure 4 (now figure 5) as suggested. The SAR group includes stramenopiles, alveolates, and rhizaria; although *Chlamydomonas* (Archaeplastida) is on the same main branch as SAR, archaeplastids are considered a separate suprakingdom (Burki *et al.*, 2020). Concerning ‘Excavates’, although the *Trypanosoma brucei* dataset is available at the EMDB (see entries 20012-20014) (Imhof *et al.* 2019), the data quality is not on par with the other organisms we presented in our manuscript, and therefore it would not make a useful addition to the revised summary figure 4 (former figure 7). Even the dataset with the best resolution (EMD-20012, listed at 21.0 Å) lacks continuously visible radial spokes, and especially the radial spoke heads – one of the key structures we are comparing – are very noisy, as shown in Author response image 1 from the EMDB website. We agree with the reviewer that sampling more eukaryotic biodiversity is important, and we have begun to investigate representatives of the ‘Excavates’ and other major branches as future studies.

**Author response image 1. sa2fig1:** 

ReferencesAdl SM, Simpson AG, Lane CE, Lukes J, Bass D, Bowser SS, Brown MW, Burki F, Dunthorn M, Hampl V et al. 2012. The revised classification of eukaryotes. *J Eukaryot Microbiol* 59: 429493.

Bloodgood RA. 2010. Sensory reception is an attribute of both primary cilia and motile cilia. *J Cell Sci* 123: 505-509.

Brunet T, King N. 2017. The Origin of Animal Multicellularity and Cell Differentiation. *Dev Cell* 43: 124-140.

Burki F. 2014. The eukaryotic tree of life from a global phylogenomic perspective. *Cold Spring Harb Perspect Biol* 6: a016147.

Carbajal-Gonzalez BI, Heuser T, Fu X, Lin J, Smith BW, Mitchell DR, Nicastro D. 2013. Conserved structural motifs in the central pair complex of eukaryotic flagella. *Cytoskeleton (Hoboken)* 70: 101-120.

Fu G, Zhao L, Dymek E, Hou Y, Song K, Phan N, Shang Z, Smith EF, Witman GB, Nicastro D. 2019. Structural organization of the C1a-e-c supercomplex within the ciliary central apparatus. *J Cell Biol* 218: 4236-4251.

Hibberd DJ. 1975. Observations on the ultrastructure of the choanoflagellate Codosiga botrytis (Ehr.) Saville-Kent with special reference to the flagellar apparatus. *J Cell Sci* 17: 191-219.

Kollmar M. 2016. Fine-Tuning Motile Cilia and Flagella: Evolution of the Dynein Motor Proteins from Plants to Humans at High Resolution. *Molecular biology and evolution* 33: 3249-3267.

Leadbeater B. 2006. The 'mystery' of the flagellar vane in choanoflagellates. *Nova Hedwigia*: 213-223.

Levin TC, King N. 2013. Evidence for sex and recombination in the choanoflagellate *Salpingoeca rosetta*. *Curr Biol* 23: 2176-2180.

Li S, Fernandez JJ, Fabritius AS, Agard DA, Winey M. 2022. Electron cryo-tomography structure of axonemal doublet microtubule from *Tetrahymena thermophila*. *Life Sci Alliance* 5.

Lin J, Heuser T, Song K, Fu X, Nicastro D. 2012. One of the nine doublet microtubules of eukaryotic flagella exhibits unique and partially conserved structures. *PLoS One* 7: e46494.

Ma M, Stoyanova M, Rademacher G, Dutcher SK, Brown A, Zhang R. 2019. Structure of the Decorated Ciliary Doublet Microtubule. *Cell* 179: 909-922 e912.

Mah JL, Christensen-Dalsgaard KK, Leys SP. 2014. Choanoflagellate and choanocyte collar-flagellar systems and the assumption of homology. *Evol Dev* 16: 25-37.

Mehl D, Reiswig HM. 1991. The presence of flagellar vanes in choanomeres of Porifera and their possible phylogenetic implications. *Journal of Zoological Systematics and Evolutionary Research* 29: 312-319.

Mitchell DR. 2007. The evolution of eukaryotic cilia and flagella as motile and sensory organelles. *Adv Exp Med Biol* 607: 130-140.

Nielsen LT, Asadzadeh SS, Dolger J, Walther JH, Kiorboe T, Andersen A. 2017. Hydrodynamics of microbial filter feeding. *Proc Natl Acad Sci U S A* 114: 9373-9378.

Pazour GJ, Agrin N, Leszyk J, Witman GB. 2005. Proteomic analysis of a eukaryotic cilium. *J Cell Biol* 170: 103-113.

Sigg MA, Menchen T, Lee C, Johnson J, Jungnickel MK, Choksi SP, Garcia G, 3rd, Busengdal H, Dougherty GW, Pennekamp P et al. 2017. Evolutionary Proteomics Uncovers Ancient Associations of Cilia with Signaling Pathways. *Dev Cell* 43: 744-762 e711.